# Matrin3 mediates differentiation through stabilizing chromatin loop-domain interactions and YY1 mediated enhancer-promoter interactions

Tianxin Liu[1,9], Qian Zhu[1,6,9], Yan Kai[1], Trevor Bingham[2], Stacy Wang[3], Hye Ji Cha[1,7], Stuti Mehta[1], Thorsten M. Schlaeger [2], Guo-Cheng Yuan [4,8] & Stuart H. Orkin [1,5] ✉

Although emerging evidence indicates that alterations in proteins within nuclear compartments elicit changes in chromosomal architecture and differentiation, the underlying mechanisms are not well understood. Here we investigate the direct role of the abundant nuclear complex protein Matrin3 (Matr3) in chromatin architecture and development in the context of myogenesis. Using an acute targeted protein degradation platform (dTAG-Matr3), we reveal the dynamics of development-related chromatin reorganization. High-throughput chromosome conformation capture (Hi-C) experiments revealed substantial chromatin loop rearrangements soon after Matr3 depletion. Notably, YY1 binding was detected, accompanied by the emergence of novel YY1-mediated enhancer-promoter loops, which occurred concurrently with changes in histone modifications and chromatin-level binding patterns. Changes in chromatin occupancy by Matr3 also correlated with these alterations. Overall, our results suggest that Matr3 mediates differentiation through stabilizing chromatin accessibility and chromatin loop-domain interactions, and highlight a conserved and direct role for Matr3 in maintenance of chromosomal architecture.

Nuclear proteins in eukaryotes comprise non-chromatin microgranular, ribonucleoproteins, connecting the nuclear membrane to intranuclear components. Acting as scaffold for attachment of chromatin, the nuclear protein complex serves to support and organize chromatin architecture. Together with lamins and heterogeneous nuclear ribonucleoproteins, Matrin3 (Matr3) is the major constituent of the inner nuclear proteins[1,2].

Attention in the literature has focused primarily on the involvement of Matr3 in pre-messenger RNA (mRNA) splicing, mRNA stability, and DNA damage repair and replication[3–5]. Mutations in Matr3 contribute

[1]Dana-Farber/Boston Children's Cancer and Blood Disorders Center, Harvard Stem Cell Institute, Harvard Medical School, Boston, MA 02115, USA. [2]Stem Cell Program, Boston Children's Hospital, Boston, MA 02115, USA. [3]Lester Sue Smith Breast Center, Department of Human Molecular Genetics, Baylor College of Medicine, 1 Moursund St, Houston, TX 77030, USA. [4]Department of Pediatric Oncology, Dana-Farber Cancer Institute and Harvard Medical School, Boston, MA 02115, USA. [5]Howard Hughes Medical Institute, Boston, MA 02115, USA. [6]Present address: Lester Sue Smith Breast Center, Department of Human Molecular Genetics, Baylor College of Medicine, 1 Moursund St, Houston, TX 77030, USA. [7]Present address: Department of Biomedical Science & Engineering, Dankook University, Cheonan 31116, South Korea. [8]Present address: Department of Genetics and Genomic Sciences, Icahn School of Medicine at Mount Sinai, New York, NY 10029, USA. [9]These authors contributed equally: Tianxin Liu, Qian Zhu. ✉e-mail: stuart_orkin@dfci.harvard.edu

to a subset of familial amyotrophic lateral sclerosis (ALS)[6,7]. Matr3 has also been implicated in developmental processes, as its knockout is embryonic lethal at or before the E8.5 neural-fold stage[8], and its deficiency promotes neural stem differentiation[9]. Recently, we reported that loss of Matr3 in embryonic stem cells and erythroid precursors elicited changes indicative of accelerated differentiation, and was associated with altered chromatin organization[10].

To explore how Matr3 participates in chromatin organization more broadly, here we have investigated its loss in the context of an established and tractable differentiation system, the myoblast cell line, C2C12[11–14]. Studying an abundant nuclear component, such as Matr3, presents challenges, as analysis of a cellular phenotype following gene knockout and accompanying compensatory changes may confound interpretation of its primary roles. To dissect the role of Matr3 in chromatin organization more directly, we have employed targeted protein degradation (TPD). Acute depletion of Matr3 by TPD afforded the opportunity to interrogate chromatin accessibility, transcription (TF) bind, and chromatin organization in a temporal fashion. Our findings identify a distinct role for Matr3 in stabilizing chromatin architecture during cellular differentiation, further establishing a critical requirement beyond its more commonly appreciated involvement in RNA metabolism.

## Results

### Matr3 loss leads to defects in myogenesis
We first evaluated the effect of *Matr3* loss on myogenesis within the context of differentiation of C2C12 cells. CRISPR/Cas9 editing was used to delete the entire *Matr3* gene body (Supplementary Fig. 1a–c). Upon analysis of individual C2C12 knockout clones, we observed clonal variation in growth and differentiation (Supplementary Fig. 1d). To circumvent clonal effects, we generated cells depleted of Matr3 in bulk using a single guide delivered as RNP (Fig. 1a, Supplementary Fig. 2). The Matr3 level was reduced by ~90% in the bulk population (Fig. 1b). Upon differentiation, myotubes depleted of Matr3 exhibited a significantly increased density than wildtype, reflective of hypertrophy (Fig. 1c, Supplementary Fig. 3). This phenotype mimics that of Duchenne muscular dystrophy (DMD), which is characterized by a high percentage of branched myofibers that are vulnerable to contraction-induced injury[15].

To assess the impact of Matr3 loss on gene expression, we performed RNA-seq of cells at myoblast (Day 0) and myotube (Day 4) stages. Changes in gene expression were observed, as reflected in enrichment of gene sets associated with skeletal muscle development (Fig. 1d). Among differential gene sets, Duchenne muscular dystrophy gene (DMD) expression and DCAF8 were reduced, which was confirmed by protein analyses (Fig. 1e-f). Thus, depletion of Matr3 leads to aberrant cell differentiation.

### Acute Matr3 depletion elicits few changes in nascent RNA production
To identify more immediate consequences of Matr3 loss on gene expression and chromatin organization, we employed PROTAC-mediated TPD using the dTAG platform[16,17]. Using CRISPR/Cas9-mediated editing, facilitated by AAV template DNA, we engineered C2C12 cells harboring variant FKBP introduced in-frame at the N-terminus of the *Matr3* gene (Fig. 2a). Leveraging high-efficiency gene editing in bulk populations, we avoided clonal variation (Fig. 2b and Supplementary Fig. 3a, b). Following exposure to PROTAC dTAG47, Matr3 protein was depleted within 4 h. (Fig. 2c, and Supplementary Fig. 4c, d).

To identify early transcriptional changes upon Matr3 loss, we performed SLAM-seq at 4 h. post dTAG47 exposure and quantified nascent RNA transcription[18]. We observed only ~8 significantly differentially expressed (DE) genes ($\log_2 FC > 0.4$) (Fig. 2d, Supplementary Fig. 5). Expanding the DE gene list further, we observed that genes at

the top of the DE gene list were highly similar to the 24 h. RNA-seq DE gene list (e.g., Eps15l1, Dcaf8, Asb3, Thumpd2, Naa25) (Supplementary Table. 1), suggesting that critical DE genes are captured by the acute depletion system. Thus, acute depletion of Matr3 (4 h.) elicited very limited changes in gene expression.

### Matr3 depletion impacts chromatin accessibility and MyoD binding
Because limited gene expression changes often belie more extensive alterations in the chromatin organization, we next tracked chromatin accessibility upon loss of Matr3[19]. In the Matr3 KO system, we observed predominantly loss of chromatin accessibility (Fig. 3a). To dissect these changes in detail, we assessed chromatin accessibility upon acute depletion of Matr3. We observed both gains and loss of chromatin accessibility at an early time (4 h.), which was followed by a trend to greater loss at 8 h and predominant loss thereafter (steady-state). We further assessed the effects of prolonged Matr3 loss on myogenesis by examining chromatin occupancy of the master regulator MyoD by CUT&RUN[20,21]. We detected a similar trend to that observed for overall chromatin accessibility. Increased MyoD binding was seen at the early time point followed by gradual loss of MyoD binding (Fig. 3b). Based on these findings, we surmise that depletion of Matr3 directly initiates chromatin changes characterized by increased accessibility and MyoD occupancy.

Upon mapping the differentially accessible regions to the genes within 25 kb of the transcription start sites (TSS) (Fig. 3c), we observed that loci with increased accessibility included MAPK kinase Map3k4, guanine nucleotide exchange factor Rapgef4, transcription factor Ctbp2, and skeletal muscle target Myh. Gene set enrichment analysis by DAVID[22] and SEEK[23] of differentially accessible genes revealed embryonic organ morphogenesis ($P = 1.7e\text{-}4$), muscle structure development ($P = 3.7e\text{-}2$), regulation of signal transduction ($P = 2.7e\text{-}2$), cell motility/cell migration at 4 h. following Matr3 depletion (up-regulated genes) (Fig. 3e).

We next asked if early changes in chromatin accessibility could be attributed to redistribution of MyoD occupancy. We mapped differential MyoD occupancy loci to nearby genes (Fig. 3d). MyoD differential binding regions were enriched in genes in extracellular matrix organization ($P = 1e\text{-}9$), cellular response to growth factor stimulus ($P = 1.7e\text{-}6$), myofibril assembly, and regulation of signal transduction ($P = 9e\text{-}4$) (Fig. 3f). Upon analyzing the overlap between differential ATAC-seq and differential MyoD binding, we observed a total overlap of 253 differential peaks, which represents a highly significant portion of total differential ATAC-seq and MyoD peaks ($P = 2e\text{-}15$) (Fig. 3g), suggesting that Matr3 depletion simultaneously perturbs open chromatin regions and MyoD binding. Collectively, these results indicate that acute depletion of Matr3 directly initiates changes in chromatin accessibility and MyoD binding at selective sites that are related to signal transduction and development.

### Gene expression changes accumulate at later developmental stages
Although acute depletion of Matr3 affected differential chromatin accessibility and MyoD chromatin occupancy, few changes in gene expression were observed at early times (Fig. 2d). Thus, we hypothesized that early chromatin changes lead to subsequent gene expression changes later in muscle development. We tracked the expression of genes that are downstream of sites with differential chromatin accessibility and MyoD occupancy upon depletion of Matr3 for 4 h. To avoid potential effects of prolonged dTAG47 treatment on cell viability and/or differentiation, we examined gene expression in bulk Matr3 KO cells at different days of differentiation (Fig. 3h, i, h for upregulated in Matr3KO, i for downregulated in Matr3KO). Genes nearby perturbed MyoD/accessibility sites exhibited significant gene expression

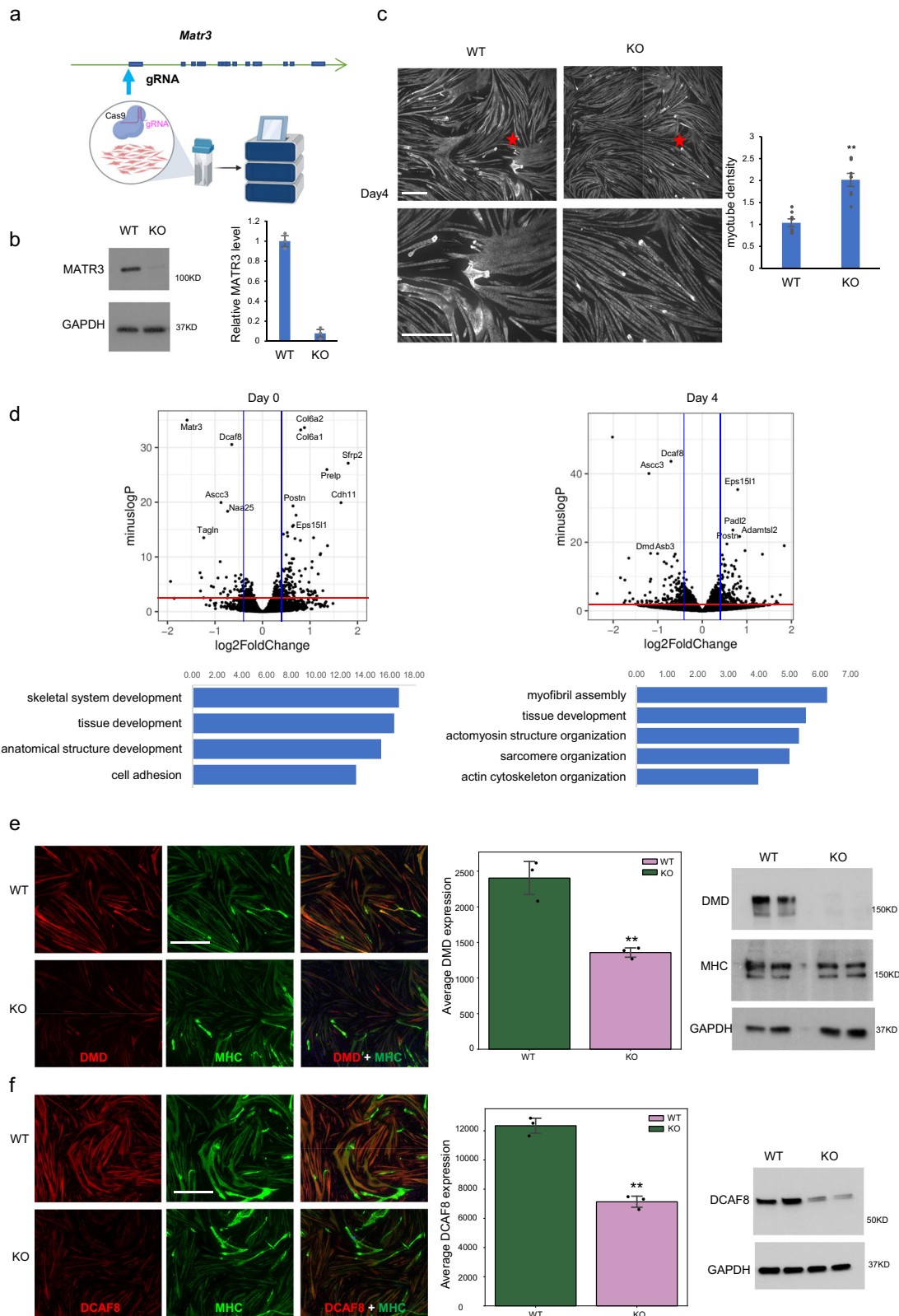

differences in 4 and 6 day differentiated Matr3 KO bulk cells (KO vs WT). No appreciable changes in expression were observed at day 0 of differentiation. Using a large-scale public data-mining approach[23], we deduced a general list of coordinated genes that were co-expressed with genes encoding the Matr3-complex[24] (Supplementary Fig. 6).

These co-expressed genes (n = 500), representing Matr3-coordinated genes, were most significantly altered in gene expression in day 6 differentiated Matr3 KO cells, indicating that targets of a Matr3-protein complex were altered at a later time by Matr3 loss. Overall, earlier changes in chromatin accessibility and MyoD differential occupancy

**Fig. 1 | Matr3 loss leads to defects in myogenesis. a** Generation of Matr3 knockout (KO) bulk using Cas9/RNP transfection (see methods). Figure created with BioRender.com. **b** Matr3 protein level was reduced significantly in Matr3KO bulk. MATR3 expression was assessed in wildtype and Matr3 KO C2C12 cells by Western blot and quantified ($n = 3$ independent experiments). $p = 1.25E\text{-}05$, by two-sided $t$ test. **c** Myotubes in Matr3 KO exhibited a significantly increased density than wildtype. C2C12 Matr3 KO and wildtype were differentiated for 4 days, and immunostained with Myosin heavy chain. The bottom images were magnified views from the regions marked with stars in upper images. Myotube intensity in Matr3KO and WT were quantified ($n = 8$ independent samples). $p = 2.96E\text{-}05$, by two-sided $t$ test. All the images were taken from the same regions in each replicate using Yokogawa CV7000 microscope with the same setting. Scale bar, 500 μm. **b, c** data are presented as mean values ±SEM. **d** Knockout of Matr3 contributed to differential gene expression. RNA-seq of cells at myoblast (Day0) and myotube (Day4) stages. Expression changes were measured by KO-WT. In volcano plots, $p$ values

from the Wald test less than 0.05 ($p < 0.05$) for red line, and $\log2FC > 0.4$ for the vertical blue lines denote significant DE genes. DE genes were associated with skeletal muscle development (GO term, bottom panel). **e** Duchenne muscular dystrophy gene (DMD) expression was reduced in Matr3 KO myotubes. C2C12 Matr3 KO and wildtype were differentiated 4 days, and immunostained with DMD (red) and MHC (green). Signal intensity was quantified ($n = 3$). Scale bar, 500 μm. $p = 0.0089$, by Mann-Whitney U test. Protein level of DMD was confirmed by Western blots (right panel), and 2 independent experiments were repeated with similar results. **f** DCAF8 expression was reduced in Matr3 KO myotubes. C2C12 Matr3 KO and wildtype were differentiated 4 days, and immunostained with DCAF8 (red) and MHC (green). Signal intensity was quantified ($n = 3$). Scale bar, 500 μm. $p = 0.00016$, by Welch's $t$ test. Protein level of DCAF8 was confirmed by Western blot (right panel), and 2 independent experiments were repeated with similar results. **e, f** data are presented as mean values ±SD. Source data are provided as a Source Data file.

consequent to Matr3 depletion foreshadowed later developmental gene expression changes.

## Matr3 depletion maintains A/B compartments but rearranges loop domain interactions at a sub-compartment level

Consistent with prior studies of ES and erythroid cells[10], depletion of Matr3 (Matr3 KO bulk) in C2C12 cells was associated with changes in CTCF and reduced overall cohesin (assessed by Rad21) occupancy (Supplementary Fig. 7a). Notably, changes in CTCF occupancy and reduced Rad21 occupancy were observed as early as 4h. post depletion of Matr3 (Fig. 4a). These findings extend our prior observations by implicating direct involvement of Matr3 in maintaining chromatin organization. To probe this aspect more fully, we performed high-throughput chromosome conformation capture (Hi-C) experiments upon acute depletion of Matr3 (4 h. dTAG47)[25]. No changes were observed in A/B compartment switching and interaction (Supplementary Fig. 7b, c). Instead, we detected extensive rearrangement of chromatin loops (Fig. 4b, highlighted by circles). We then mapped the rearranged loops on all chromosomes (Fig. 4c). Rearranged loops were concentrated in specific genomic, notably within Chr7, Chr14 and ChrX (Fig. 4d). As an example, we highlight Tgfb1-Ltbp4 loop rearrangements, which were ranked among the most re-arranged regions on chromosome 7 (Fig. 4e). Both gained and lost loops were adjacently located at the intersection point (see circle in Fig. 4f). Together, these findings indicate that Matr3 loss extensively perturbs chromatin loop formation.

To evaluate the effects of Matr3 depletion on chromatin architecture at steady-state, we performed Hi-C using Matr3 KO bulk on Day 0. Indeed, rearranged loops were observed upon Matr3 loss (Supplementary Fig. 8). Moreover, changes in chromatin loops were correlated with differential open chromatin accessibility (Supplementary Fig. 9). Taken together, these results provide strong evidence that Matr3 is essential in maintenance of chromatin loops around chromatin accessible regions of the genome.

## Loss of Matr3 disrupts YY1 binding and YY1-enriched cohesin loading

Chromatin loops are comprised of structural chromatin loops, and, on a finer scale, long-range chromatin interactions, including enhancer-enhancer (E-E), enhancer-promoter (E-P) and promoter-promoter (P-P) loops[26–28]. The strength of E-P and P-P interactions positively correlates with the level of gene expression[29]. Yin Yang 1, YY1, a ubiquitously expressed transcription factor that preferentially occupies enhancers and promoters in mammalian cells, is a critical regulator of E-P loops, whereas CTCF is enriched at insulator elements[30,31]. Matr3 was found in complexes with CTCF and cohesin; moreover, CTCF and cohesin occupancy was decreased upon Matr3 loss[10]. Given the reported association between YY1 and cohesin[29,30], we hypothesized that Matr3

might impact chromosomal loops by affecting cohesin and YY1 occupancy.

By CUT&RUN analysis (4 h. dTAG47 treatment), we observed that YY1 binding was enhanced at 2840 locations (compared to 947 locations where YY1 was decreased) upon Matr3 depletion (Fig. 5a). To interrogate potential interactions between YY1 and cohesin Rad21, we mapped the pattern of redistribution of Rad21 binding after Matr3 depletion (4 h. dTAG47). Acute removal of Matr3 was associated with reduced Rad21 occupancy adjacent to enhanced YY1 binding sites (Fig. 5b, c). This pattern of enhanced YY1 binding and loss of Rad21 binding was observed at 208 genomic loci, corresponding to roughly 19% of sites at which Rad21 binding was lost (1127 sites). These observations suggest that loss of cohesin Rad21 following depletion of Matr3 might be compensated in part by increased YY1 occupancy. In addition to decreased Rad21 occupancy, sites of differential YY1 binding were often found in association with increased MyoD occupancy (Fig. 5d) and loss of chromatin accessibility (Fig. 5e). To investigate the link between increased YY1 binding and MyoD recruitment, we examined MyoD occupancy after single YY1 knockout and double Matr3 and YY1 knockout (Supplementary Fig. 10). We found that in the absence of YY1 and Matr3, the number of MyoD occupied sites decreased, implying early recruitment of MyoD is dependent on YY1. We suggest, therefore, that Matr3 loss perturbs connections between the transcriptional machinery and the higher order chromatin structure in part through changes in YY1 binding.

## Matr3 loss disrupts YY1-mediated enhancer-promoter loop formation

As YY1 is a primary regulator of EP loops[30], we hypothesized that loss of Matr3 would elicit changes in the chromatin loop domains. We next investigated YY1-mediated chromatin loop changes upon depletion of Matr3 (4 h., dTAG47). Loci characterized by enhanced YY1 occupancy exhibited a greater number of increased E-E and E-P loops. The Mphosph8 locus illustrates these changes (Fig. 6a). Increased interactions were observed between segment 1 and segments 2, 3, 4, while decreased interactions were detected only at segment 5. In the entire 1.5 Mb region, we detected 11 increased interactions as compared with 4 interactions that were decreased.

Genome-wide, increased looping upon Matr3 depletion was observed more frequently where increased YY1 occupancy was detected at the enhancer (E) anchor of differential E-P loops, marked by H3K27Ac. Furthermore, enhancers marked by increased YY1 and altered H3K27Ac deposition exhibited a higher percentage of gained interactions than lost interactions (the gain/loss ratio was 196:146, bias: 1.36, see Fig. 6b, ΔYY1&ΔH3K27Ac). This bias towards gained interactions was much stronger than gained YY1 sites with unaltered H3K27Ac (the gain/loss ratio was 1368:1246, bias: 1.09, see ΔYY1&H3K27Ac), or all H3K27Ac sites (the gain/loss ratio was 11312:10430, bias: 1.06, see

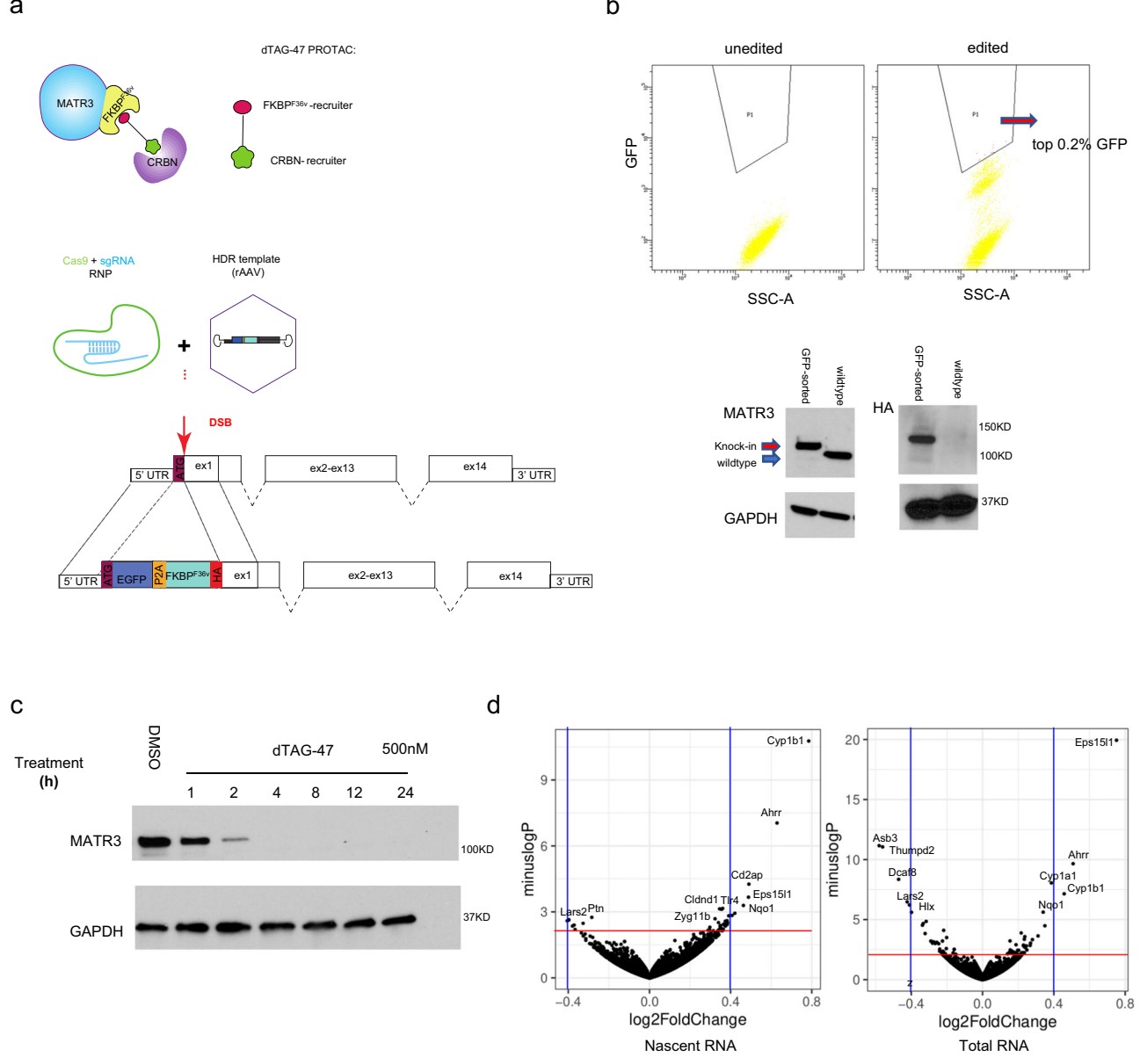

**Fig. 2 | dTAG-Matr3 could be degraded within 4 h., and acute depletion of Matr3 contributes to few changes in nascent RNA production. a** Strategy to knockin FKBP[F36v] at the 5' end of Matr3. The dTAG-47 PROTAC recruits the CRBN E3 ligase complex to FKBP[F36v]-Matr3. **b** High-efficient knockin bulk FKBP[F36v]-Matr3 were sorted by GFP and confirmed by Western blot. Cells enriched in the top 0.2% highest GFP signal were sorted (GFP-sorted). FKBP[F36v]-Matr3 (refer as dTAG-Matr3) bands were enriched in the GFP-sorted bulk, and tagging efficiency was also confirmed by HA antibody. The confirmation of Knockin was repeated 3 times independently with similar results. **c** Matr3 protein was depleted within 4 h. upon dTAG47 exposure. Western blots of dTAG-Matr3 upon dTAG47 (500 nM) treatment in a time course. Confirmation of degradation was repeated 3 times independently with similar results. **d** Nascent RNA and total RNA were quantified by SLAM-seq upon 4 h. Matr3 depletion. In volcano plots, *p* values from the Wald test less than 0.05 ($p < 0.05$) for red line, and log2FC > 0.4 for the vertical blue lines denote significant DE genes. Source data are provided as a Source Data file.

H3K27Ac). Thus, Matr3 depletion is associated with gained E-P loops at enhancer YY1 binding sites marked by H3K27Ac deposition.

An alternative scenario was observed in which E-P loops coincided with changes in active marks H3K4me3 at gene promoters. We analyzed the E-P loops of this category of genes, where H3K4me3 was gained at the gene promoter and enhancers were occupied by YY1 (Fig. 6c). We observed more balanced E-P loop changes radiating from such promoters (gain/loss ratio is 1.01) (see Δyy1e&Δh3k4me3 compared with Δyy1e&h3k4me3 in Fig. 6c). This reduction of E-P gains at differential H3K4me3 sites may reflect greater direct promoter-based regulation than E-P loop-based gene regulation. Taken together, these data reflect diverse H3K27Ac-based and H3K4me3-based mechanisms

by which Matr3 depletion elicited changes in the YY1-mediated E-P loop landscape.

Given that changes in MyoD and Rad21 occupancy, and in chromatin accessibility, as assessed by ATACseq, appeared to be strongly associated with YY1, we hypothesized that there might be a subset of binding sites with co-occurring changes most associated with such interactions. We computed genome-wide the number of loci characterized by 4-way differential chromatin accessibility and MyoD, Rad21, and YY1 binding, and the number of interaction differences at these locations, as revealed by Hi-C. Overall, we identified 17 loci that have interaction differences. 3 loci were prioritized on the basis of most altered interactions (Fig. 6d). Among the candidates,

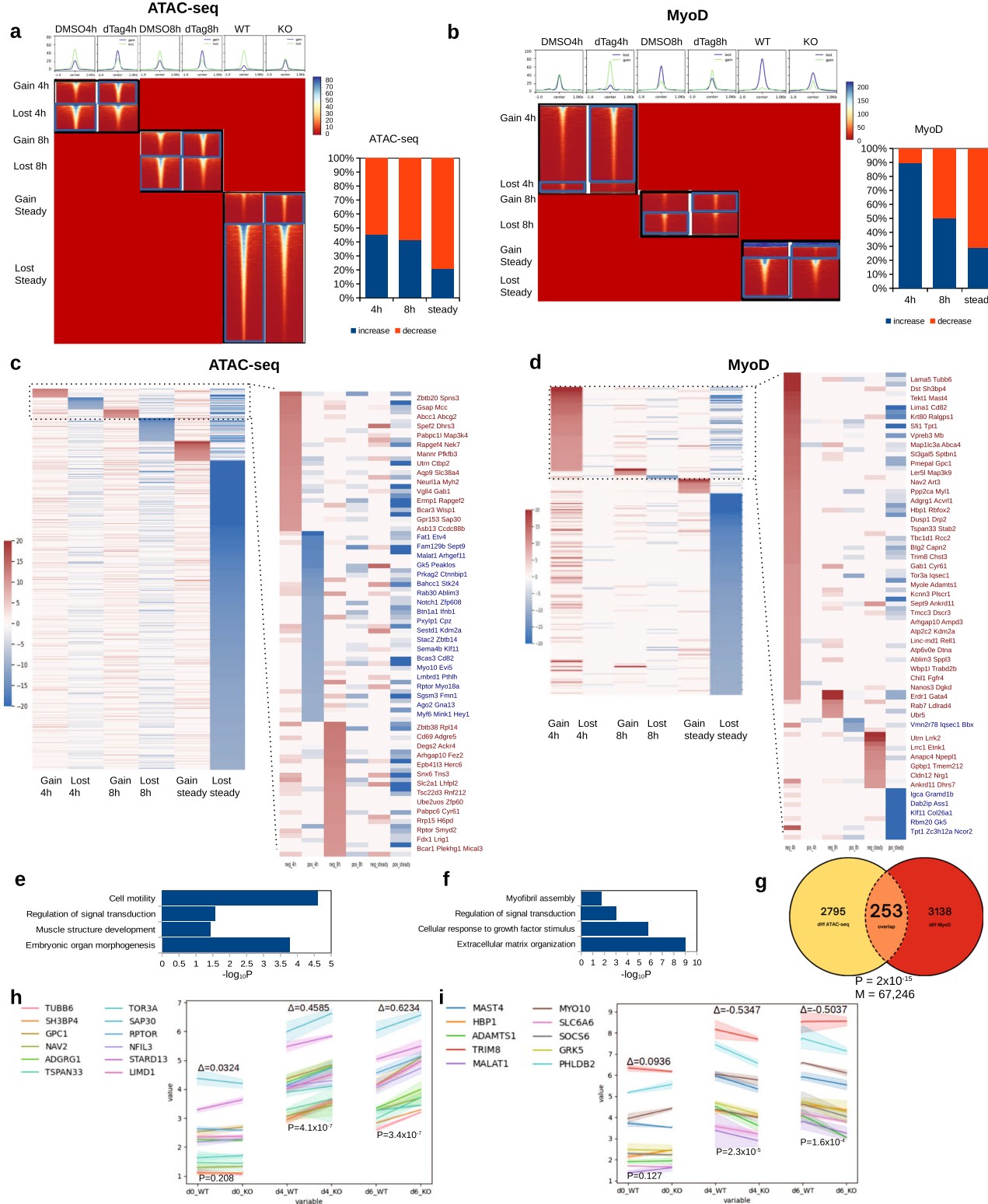

Cd82 and Kpna1 were notable for their involvement in muscle stem cell activation and proliferation with links to skeletal muscle defects[32,33]. Depletion of Kpna1 was associated with premature activation of muscle stem cells, and subsequent proliferation, apoptosis, and satellite cell exhaustion[33]. Indeed, the top 3 genes (Kpna1, Cd82, Gnb1l) (Fig. 6d) that exhibit 4-way co-differential marks along with strong loop interaction changes all displayed differential gene expression ($P = 0.0531$, $P = 0.054$, $P = 0.0029$ respectively, Day 0).

## Matr3 depletion directly alters TF occupancy and chromatin loops

To understand how loss of Matr3 leads to changes in MyoD and YY1 occupancy, and chromatin loop organization, we mapped Matr3 occupancy by CUT&RUN. In wild-type cells (DMSO), Matr3 CUT&RUN occupancy overlaps significantly with that of YY1 (40.9%), CTCF (33.4%), Rad21 (27.5%), and MyoD (23.6%) (Fig. 7a). These findings recapitulate co-occupancy between Matr3 and CTCF that was recently reported[34]. Upon degradation of Matr3, differential sites in CTCF,

**Fig. 3 | Matr3 depletion alters chromatin accessibility and MyoD binding, which foreshadow later gene expression changes. a** Heatmap showing differential chromatin accessibility (ATAC-seq) at 4 h., 8 h. post Matr3 depletion and steady state (Matr3 KO). Bar chart: percentage of peak increased and decreased per time point. **b** Same panels with (**a**) for MyoD binding (CUT&RUN). Gains trended down over time while losses increased. **c** Genes mapped by differentially chromatin accessible peaks per time point (4 h., 8 h., steady), illustrating the magnitude and number of changes. Accessibility was primarily lost at steady state. **d** Genes mapped by differential MyoD binding. Particularly striking were MyoD binding gains at 4 h. and loss at steady state. **e** GO biological process enrichment on differentially ATAC-seq mapped genes (4 h.). **f** GO enrichment on differential MyoD bound genes (4 h.). In (**e**, **f**), hypergeometric test, two-sided were used and Padj values are shown. Adjusted for multiple comparisons by Benjamini and Hochberg (BH) method.

**g** Overlap of differential ATAC and MyoD bound regions (4 h.). P: *P* value. Hypergeometric test (one-sided) with multiple comparisons by BH method. M: number of co-bound peaks. Number in circle: # of differential peaks. **h, i** Early differentially accessible and MyoD bound genes elicited expression changes later in development (days 4 and 6). Genes were selected from heatmap in (**c**, **d**) as having MyoD 4 h. binding gain or ATAC 4 h. gain/loss. Expression changes (measured by Δ=KO-WT) were followed during development. Expression changes were not significant at day 0 (steady state, both up and down $p > 0.1$), but evident at days 4, and 6 (both up and down $p < 0.00016$). Upregulated genes in Matr3 downregulated genes exhibited positive Δ (**h**), downregulated genes in Matr3KO show negative Δ (**i**). Error bars and bands represent 95% confidence interval of mean. Paired *t* test, one-sided was used. No multiple comparison is made. Source data are provided as a Source Data file.

Rad21, YY1, MyoD, chromatin accessibility overlap Matr3 occupancy (Fig. 7b–e). We note that YY1 and MyoD binding sites were more affected by Matr3 depletion, as the magnitude of Matr3 occupancy reduction was greatest at YY1 and MyoD binding sites (Fig. 7d, right histograms). In contrast, Matr3 occupancy was less prominent at sites of CTCF occupancy, and the majority of Matr3-CTCF co-occupied regions that changed were weak peaks (Fig. 7c, right histograms). These results are consistent with the finding that Matr3 modulates transcription through effects on YY1 occupancy. Taken together, these findings support a direct contribution of Matr3 occupancy to changes in chromatin occupancy of other factors.

To investigate further the role of Matr3 on chromatin loop arrangement, we analyzed the loops with anchors that were occupied by Matr3. Genomic loci (TSS) that display Matr3 alterations (by Matr3 CUT&RUN) were more enriched for loop rearrangements than random TSS loci (Fig. 7f, left). Interestingly, TTS (transcription-termination sites) affected by Matr3 depletion were re-arranged as well (Fig. 7f, right). Moreover, E-E and E-P loop anchors that exhibit Matr3 occupancy at anchors were more likely to be gained after Matr3 depletion, and harbor loop gain/loss imbalance characteristic of loop rewiring (Fig. 7g), suggesting possibly transcriptional activation at these genes. As an example, the Mphosph8 locus illustrates the occupancy of Matr3 at the region enriched with chromatin loop re-arrangement (Fig. 6a).

Taken together, we propose a model to account for how loss of Matr3 affects gene expression and differentiation (Fig. 8). We posit that Matr3 loss destabilizes cohesin-CTCF complex binding by reducing cohesin loading, thereby altering chromatin structural loops and affecting long distance interactions. As a mechanism for partial compensation of cohesin loss, YY1 is recruited to chromatin sites, thereby disrupting the E-P loop landscape. During myogenesis, gain of YY1-mediated E-P loops may contribute to recruitment of MyoD, leading to gene expression changes as skeletal muscle development progresses.

## Discussion

Matr3 has been traditionally ascribed with roles in RNA splicing and transport[3,5]. Our prior work demonstrates that Matr3 also associates with chromatin through interactions with architectural proteins CTCF and cohesin/Rad21, and orchestrates developmental transitions in embryonic stem cells and erythroid cells[10]. Here, through studies of Matr3 contributions in the context of myogenesis, we extend a conserved role for Matr3 in differentiation and explore in detail how its loss impacts chromosome looping.

With the goal of probing the direct roles of Matr3 in chromatin structure, rather than secondary effects resulting from its absence, we established an experimental platform for acute TPD of Matr3 in myogenic C2C12 cells by introducing a modified FKBP cassette at high efficiency into the Matr3 locus. Following addition of the PROTAC dTAG47, Matr3 was depleted within 4 h. in bulk cell populations. This strategy permitted direct assessment of the consequences of Matr3 loss on dynamic aspects of chromatin organization. Several new perspectives on the role of Matr3 arose from these studies.

First, upon TPD of Matr3 in C2C12 cells, we uncovered dynamic, early changes in overall chromatin accessibility and MyoD occupancy. Specifically, we observed mixed early gains in chromatin accessibility and predominant gains in MyoD occupancy, which were followed by loss of both accessibility and MyoD occupancy at later times (Fig. 3a, b). These early changes in chromatin accessibility and MyoD occupancy were not reflected in rapid changes in gene expression, but were associated with changes later in muscle differentiation (Fig. 3h, i). We propose that early changes in MyoD occupancy and chromatin accessibility (Matr3 4 h. TPD) represent initial steps that perturb myogenesis, which is compounded by accumulated changes in gene expression.

Second, our findings revealed that gene expression changes are manifested days later than the initial perturbation of chromatin structure elicited by TPD of Matr3. This observation is in accord with a recent study, which demonstrated that an RNA-binding protein complex containing MATR3 assembled at an inactive X (Xi)-compartment mediated gene repression even after day 3 of differentiation[24], at which point the Xist RNA had disappeared. Thus, a seeded protein condensate may have long lasting and sustained effects on gene expression[24]. In our study, we observed that genes co-expressed and found in a Matr3-complex were cumulatively perturbed upon TPD of Matr3 and subsequent C2C12 cell differentiation (Supplementary Fig. 6). These findings highlight how direct effects on chromatin architecture may only be appreciated at later times in cell differentiation.

Third, chromosomal loop rearrangements observed following TPD of Matr3 were broad in scale, reflecting a general rather than locus-specific role of Matr3 in maintaining chromatin architecture. Rearrangements included simultaneous gains of new enhancer-promoter, enhancer-enhancer loops, and losses of adjacent existing loops (Fig. 4b).

We hypothesized that loop rearrangements upon Matr3 loop are facilitated in part by the sliding of cohesin rings and YY1 occupancy near enhancer and promoter sites. We observed that acute Matr3-depletion preferentially affected sites at which both cohesin and YY1 were bound (Fig. 5b, c), and altered YY1-mediated EP loops in a histone H3K27Ac and H3K4me3-dependent manner (Fig. 6b, c). Sites of differential H3K27Ac and H3K4me3 marking harbored more EP-loop changes. Previously, YY1 was reported to directly regulate enhancer-promoter interactions[30]. Our data provide evidence that regions of YY1-binding perturbed upon Matr3 loss exhibit E-P interaction differences. Given the observed Matr3 occupied sites overlap with sites of differential MyoD/Rad21/YY1 occupancy and chromatin accessibility, and loop re-arrangement (Fig. 7), we speculate that Matr3 stabilizes E-P loops through modulation of YY1 occupancy. Upon rapid TPD of Matr3, YY1 occupancy leads to recruitment of MyoD, and differential histone acetylation to establish EP loops and activate downstream gene targets (Fig. 8). On the other hand, RNA binding may be involved in this process. Matr3 binds to RNAs[5], and it has been reported that Matr3 associates with antisense LINE1 (AS L1) RNAs[35,36], and forms a meshwork that gathers chromatin in the nucleus, and affects higher-order chromatin organization. Therefore, it is possible that Matr3 stabilizes chromatin loops in association with RNA.

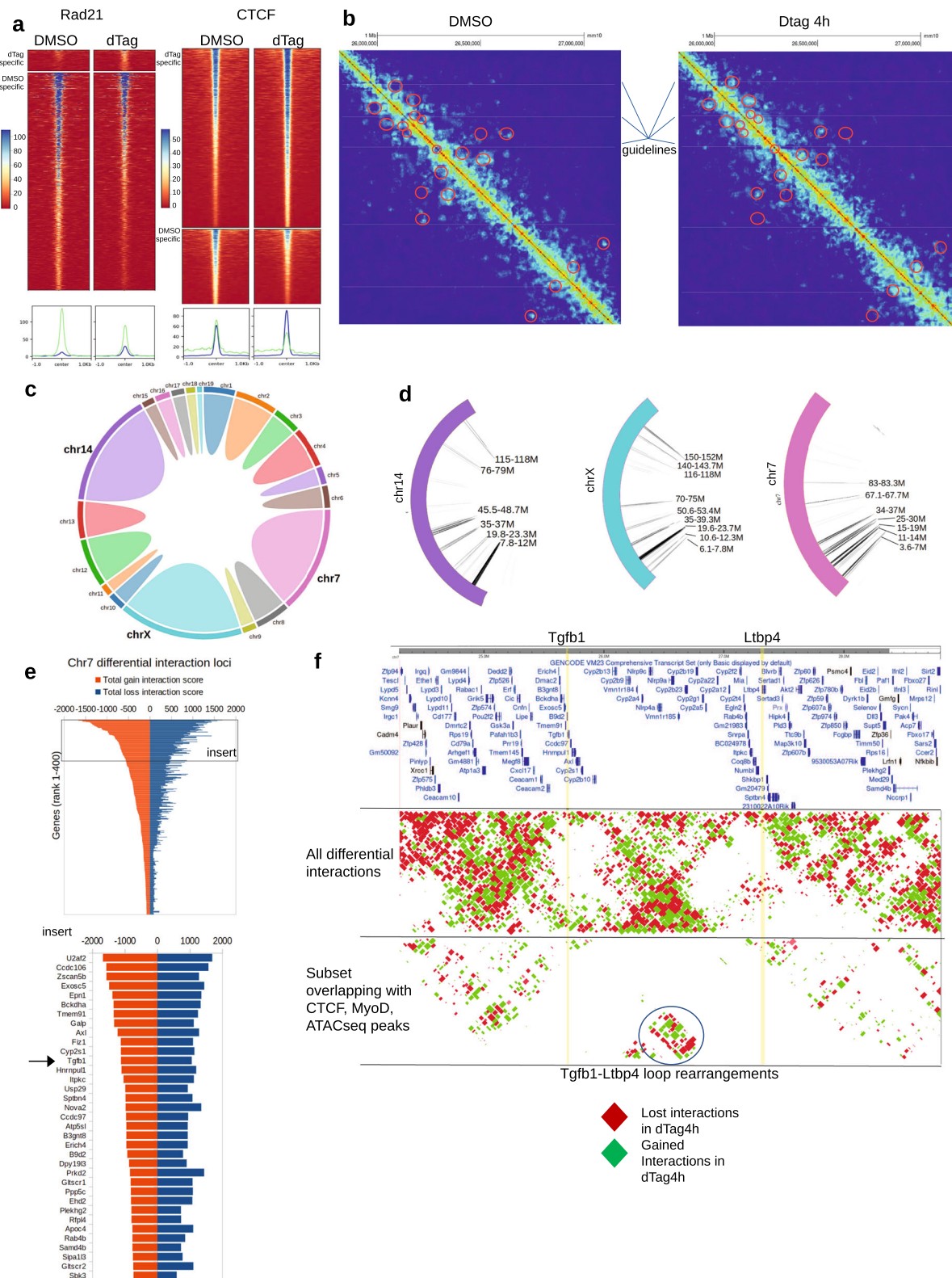

Matr3 has been reported to interact with members of the spliceosome complex and serve a role in RNA splicing[4]. Perturbation of the spliceosome member Hnrnpm led to reduced Matr3, and knockdown of AKAP8, a member of the Hnrnpm RNA splicing complex, was associated with a mesenchymal phenotype and accelerated EMT[37,38]. These findings indicate that perturbing a member of the RNA splicing complex may have developmental consequences at later times. Taken together with our observations of the role of Matr3 in maintenance of

chromatin architecture, it would be interesting to explore whether depletion of other members of the spliceosome complex elicit changes in chromatin structure and accessibility.

Mutations and dysregulation of Matr3 have been associated with ALS and Spinal Muscular Atrophy (SMA)[39]. While it has been generally accepted that these effects reflect perturbation of processes related to RNA processing/splicing[40,41], our findings implicate chromatin loop level changes as an alternative, or contributing, mechanism. We

**Fig. 4 | Matr3 depletion extensively rearranges loop domain interactions.**
**a** Substantial reduction of Rad21 binding 4 h. following Matr3 depletion. CTCF occupancy was marked by both gains and losses. The histograms below summarize the binding intensities (green: lost peaks, blue: gained peaks). **b** Example of loop rearrangements derived from Hi-C experiments that characterized the effect on chromatin structure at fragment resolution (on average 5 kb). Red circles denote sites of differential interactions. **c** Distribution of the number of differential interactions across all chromosomes. **d** Chromosomes 14, X, and 7 exhibited the greatest interaction changes with most rearranged regions indicated. **e** A ranking of

gene promoters with most differential interactions on Chromosome 7, in which Tgfb1 was ranked near the top (see insert). **f** Interaction landscape at the Tgfb1 region on Chromosome 7 revealed interweaving patterns of interaction gains (green) and losses (red). Certain sites harbored more interaction changes than others. Notable was Tgfb1-Ltbp4 loop rearrangement (circled) which also overlapped with CTCF, MyoD, ATACseq peaks. Rearrangement was formed between Tgfb1 (highlighted by yellow line), and Ltbp4 (yellow line) and bore the signature of adjacent gains and losses at the same location. Source data are provided as a Source Data file.

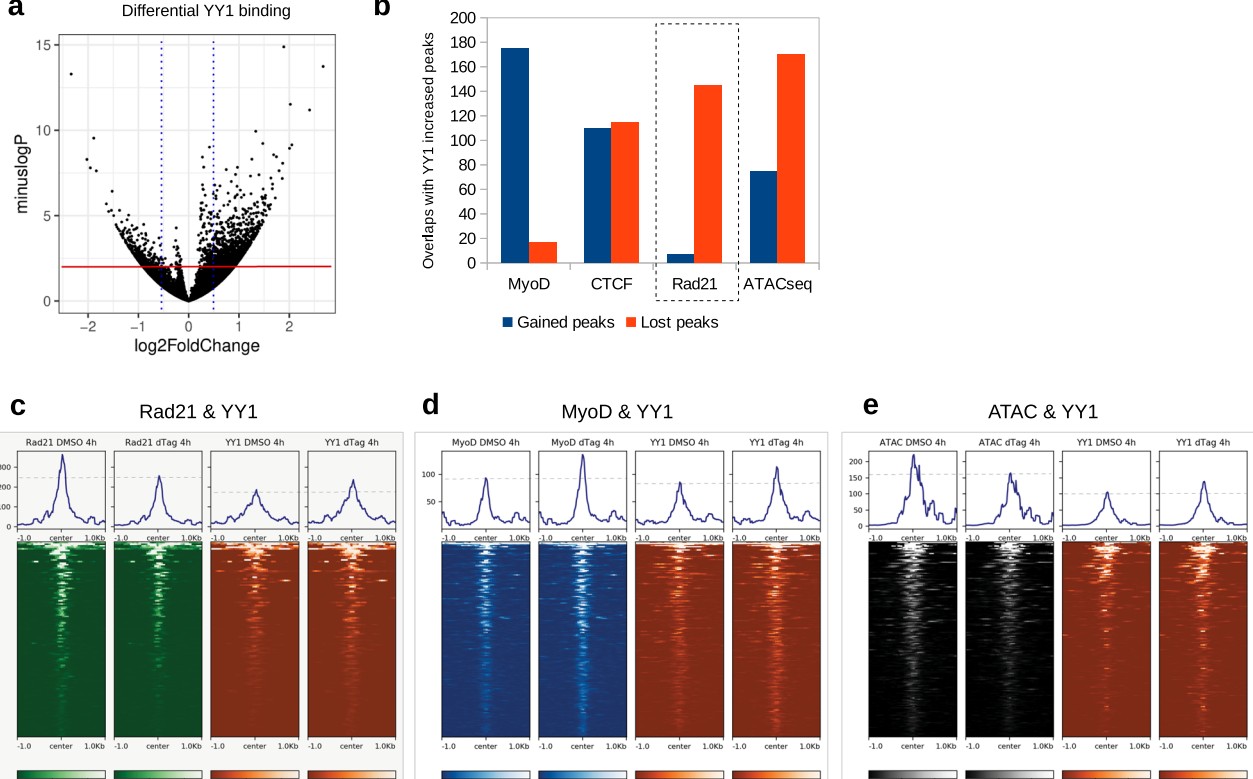

**Fig. 5 | Loss of Matr3 disrupts YY1 binding and YY1-enriched cohesin loading.**
**a** YY1 binding enhanced upon Matr3 depletion (4 h.). Volcano plot showing the logFoldChange vs. -log *P* value of YY1 binding changes (measured by Δ=dTAG-DMSO treatment). Red line: significance line (*P* = 0.05). Wald test are corrected for multiple comparison using the BH method. **b** Co-occurrence of differential YY1 occupancy with MyoD, CTCF, Rad21, ATAC-seq. Number of shared differential peaks between YY1 increased (4 h.) and the gained/lost portion (4 h.) of MyoD,

CTCF, Rad21, or ATAC-seq. Box: the overlap percentage with Rad21 binding loss was highest, showing a relationship between YY1 differential binding and cohesin loss. **c** Heatmap showing that Rad21 binding was lost at sites of YY1 binding increase. Dashed line on the histogram shows the reference line to aid comparison between dTAG47 treatment and DMSO. **d** Heatmap showing MyoD binding was enhanced at sites of YY1 binding increase. **e** Heatmap showing ATAC-seq peaks were decreased at sites of YY1 binding increase. Source data are provided as a Source Data file.

collected ~ 500 genes located in rearrangement hotspots (Supplementary Fig. 11) in ch 14, 7, X. The X chromosome is highly syntenic between mouse and human. Cytoband Chr19q13 is the human syntenic segment for chr7 in mice and contains both cancer-related and muscle disease-related genes (DMPK, DMWD, SIX5 related to Myotonic Dystrophy Type 1). For example, upon Matr3 depletion we observed rearrangement within the Ltbp4-Tgfb1 locus (Fig. 4f), a region linked to muscular dystrophy in GWAS[42–44]. Thus, we propose that the loss of Matr3 destabilizes chromatin loops at conserved hotspots related to muscular disease, and loop rearrangements contribute to changes in transcription factor occupancy and gene expression, leading to defects in development. Future efforts may be directed to investigate the significance of regulatory elements involved in long-range chromatin interactions which are perturbed upon Matr3 loss. The list of chromatin loop changes prioritized by Hi-C upon TPD of Matr3 may be

used to reinterpret susceptibility loci from neuromuscular disease GWAS studies.

In summary, our work reveals a critical role of Matr3 in development in several contexts, and directly links the alteration in proteins within nuclear compartments to changes in chromosomal architecture. Through effects on chromatin accessibility, and the occupancy of master transcriptional regulators and YY1, Matr3 impacts chromatin structure to orchestrate differentiation.

## Methods

### C2C12 Cell culturing and differentiation assay
The C2C12 cell line was obtained from the American Type Culture Collection (ATCC® CRL1772™). C2C12 cells were grown and differentiated following the standard protocol[12]. C2C12 myoblasts were propagated at 37 °C, 5% $CO_2$ in Dulbecco's modified Eagle's medium

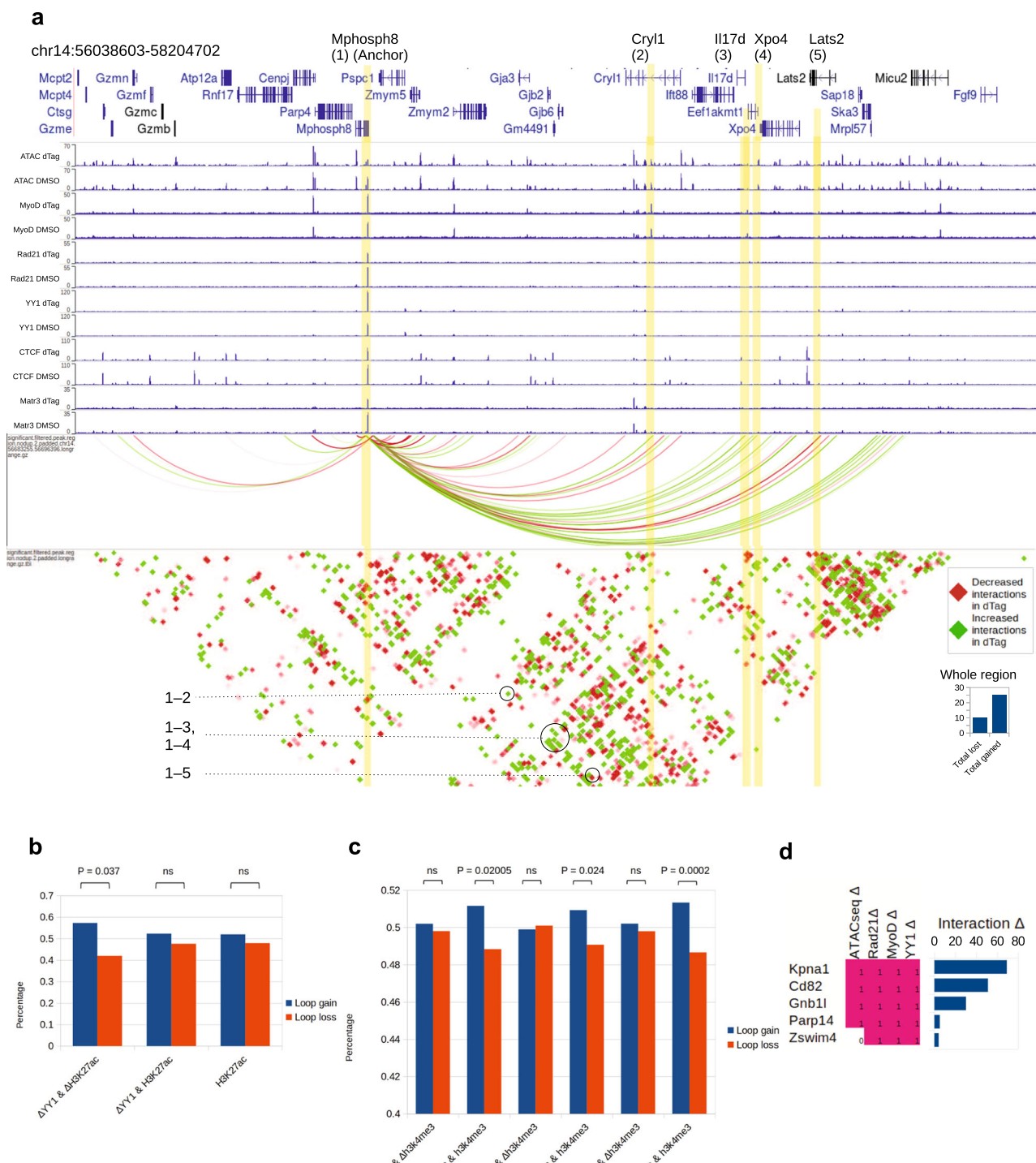

**Fig. 6 | Loss of Matr3 disrupts YY1-mediated enhancer-promoter loop formation. a** Loci characterized by enhanced YY1 occupancy exhibited increased E-E and E-P loops. Mphosph8 locus was an example of a promoter with differential YY1 binding causing EP- loop gains with neighboring MyoD, ATAC-seq peaks. Mphosph8 (highlighted by yellow line) exhibited a dual YY1-gain and Rad21-loss signature. The gene formed increased interactions (highlighted by green arcs) with Cryl1 (2), Il17d (3), Xpo4 (4), and Lats2 (5) (highlighted by yellow lines), which displayed MyoD and ATAC peaks at these positions. Interaction dot map further reaffirmed the positive increase in EP-loop interactions. Histograms (bottom right) indicated the interaction gain/loss scores across the whole region in the plot. **b** YY1-mediated EP loops changed in a H3K27ac-dependent manner in the absence of

Matr3. Loop gain/loss ratios were measured. YY1 enhancers with joint H3K27Ac change (ΔYY1 & ΔH3K27Ac) were more likely to demonstrate loop gains than YY1 enhancers with no H3K27Ac changes (ΔYY1 & H3K27Ac). **c** Increased YY1 and differential H3K4me3 sites were associated with reduction of E-P loop gain. YY1-mediated EP loops changed in a H3K4me3-dependent manner in the absence of Matr3. YY1p: YY1 promoters. YY1e: YY1 enhancers. YY1 promoters with co-differential H3K4me3 (ΔYY1p & ΔH3K4me3) were compared to YY1 promoters with no H3K4me3 changes (ΔYY1p & H3K4me3). In (**b**, **c**) Wilcoxon's rank-sum test (one-sided), corrected for multiple comparisons by BH method. **d** Sites of 4-way (ATAC-seq, Rad21, MyoD, YY1) co-differential signatures were more likely to harbor loop interaction changes. Source data are provided as a Source Data file.

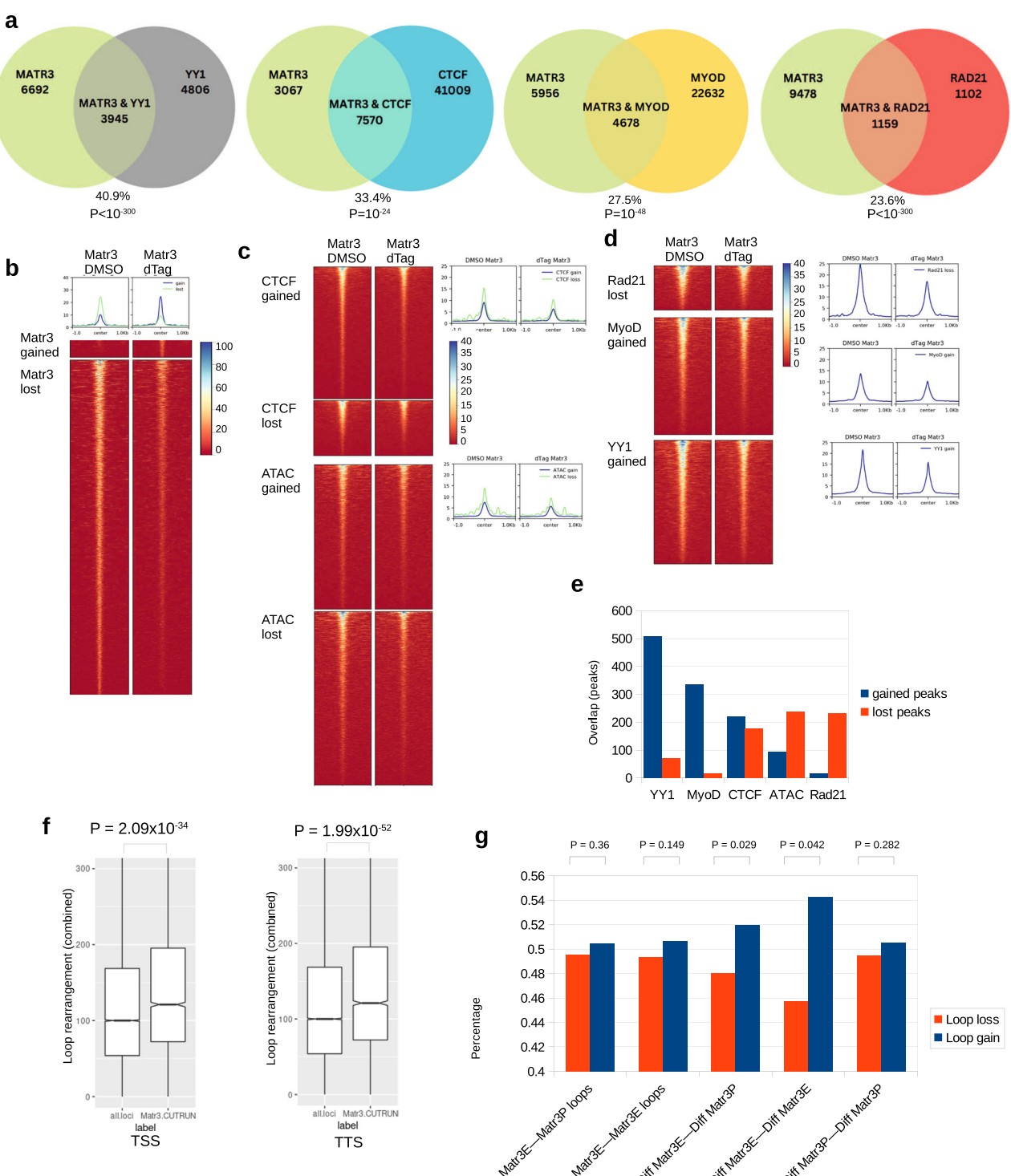

(Thermo Fisher Scientific) supplemented with 10% fetal bovine serum (Omega Scientific) and antibiotics (penicillin 100 U/ml, streptomycin 100 μg/ml, Thermo Fisher Scientific).

For differentiation, high-density cultures (~80% confluency) were switched to differentiation media (DMEM, 2% horse serum (HS, Gibco®16050–122), penicillin 100 μ/ml, and streptomycin 100 μg/ml) to induce myotube formation. Myotubes at different times (0, 2, 4, 6 days) post differentiation were used for further analysis.

**Generation Matr3 KO clones by CRISPR/Cas9**

CRISPR/Cas9 editing was used to delete the entire Matr3 gene body in C2C12 cells[10,45]. Briefly, paired single guide RNAs (sgRNAs) for site-specific cleavage of genomic regions were designed upstream and downstream of Matr3 coding region. Annealed oligos were ligated into the pX330 vector using a Golden Gate assembly strategy. A pair of CRISPR/Cas9 constructs (5 μg each) targeting each flanking region of the deletion site was transfected into $5 \times 10^5$ cells with 0.5 μg of GFP expression plasmid using BTX Harvard Apparatus. The top ~0.2% of GFP positive cells were sorted by FACS 48 hours post-transfection. Single cell derived clones were isolated and screened for biallelic deletion of target genomic sequences.

The gRNA sequences and the primers used for genotyping the clones were listed in Supplementary Table of primers and antibodies.

**Fig. 7 | Matr3 depletion directly impacts Rad21, CTCF, MyoD and YY1 occupancy, chromatin accessibility, and chromatin looping. a** Overlap between Matr3 occupancy (profiled by CUT&RUN) and YY1, CTCF, MyoD, and Rad21 occupancy. Hypergeometric test (one-sided) with multiple comparisons by BH method. **b** Heatmap showing Matr3 occupancy (CUT&RUN) at 4 h. post Matr3 depletion (dTAG) compared with control (DMSO). **c** Heatmaps showing regions with reduced Matr3 occupancy exhibit differential CTCF occupancy and differential chromatin accessibility. Right: histograms showing Matr3 CUT&RUN signal magnitude. **d** Heatmaps showing regions with reduced Matr3 occupancy exhibit lost Rad21, gained Myod, gained YY1 occupancies. Right: histograms showing Matr3 CUT&RUN signal magnitude. Note that Matr3 signals at Rad21, MyoD, YY1 locations are stronger than those of CTCF and ATAC-seq in (**c**). **e** Co-occurrence of differential Matr3 occupancy with YY1, MyoD, CTCF, Rad21 occupancy and ATAC-seq. Number of shared differential peaks between Matr3 loss with YY1 gains was highest. **f** TSS (transcription start sites) and TTS (transcription-termination sites) that occupied by Matr3 are more enriched for loop rearrangements than random loci. $n = 3$

biologically independent experiments. For TSS boxplot: $N = 16990$ (all.loci), $N = 1993$ (Matr3.CR). For TTS boxplot: $N = 16990$ (all.loci), $N = 1098$ (Matr3.CR). Boxplot shows the median, 3rd quartile (upper-bound), and 1st quartile (lower-bound). Notches in the boxplot extend the range of median $\pm 1.58 * IQR/sqrt(n)$ where IQR is the inter-quartile range. The whiskers extend no further than $1.5 * IQR$ from the hinge, and no lower than $1.5*IQR$ of the bottom hinge. Welch's $t$ test (unequal variance), no multiple comparison. **g** Different scenarios of E-P & E-E loop gains and loss at Matr3-occupied anchors. Anchors characterized by reduced Matr3 occupancy exhibited increased E-E and E-P loops. Matr3E−Matr3P loops, E-P loop anchors with no Matr3 occupancy change (1st dataset). Matr3E−Matr3E loops, E-E loops with no Matr3 occupancy change (2nd dataset). Diff Matr3E−Diff Matr3P, E-P loop with reduced Matr3 occupancy (3rd dataset, $p = 0.029$). Diff Matr3E−Diff Matr3E, E-E loop with reduced Matr3 occupancy (4th dataset, $p = 0.042$). Diff Matr3P−Diff Matr3P, P-P loop with reduced Matr3 occupancy (5th dataset). Wilcoxon's rank-sum test (one-sided), multiple comparisons by BH method. Source data are provided as a Source Data file.

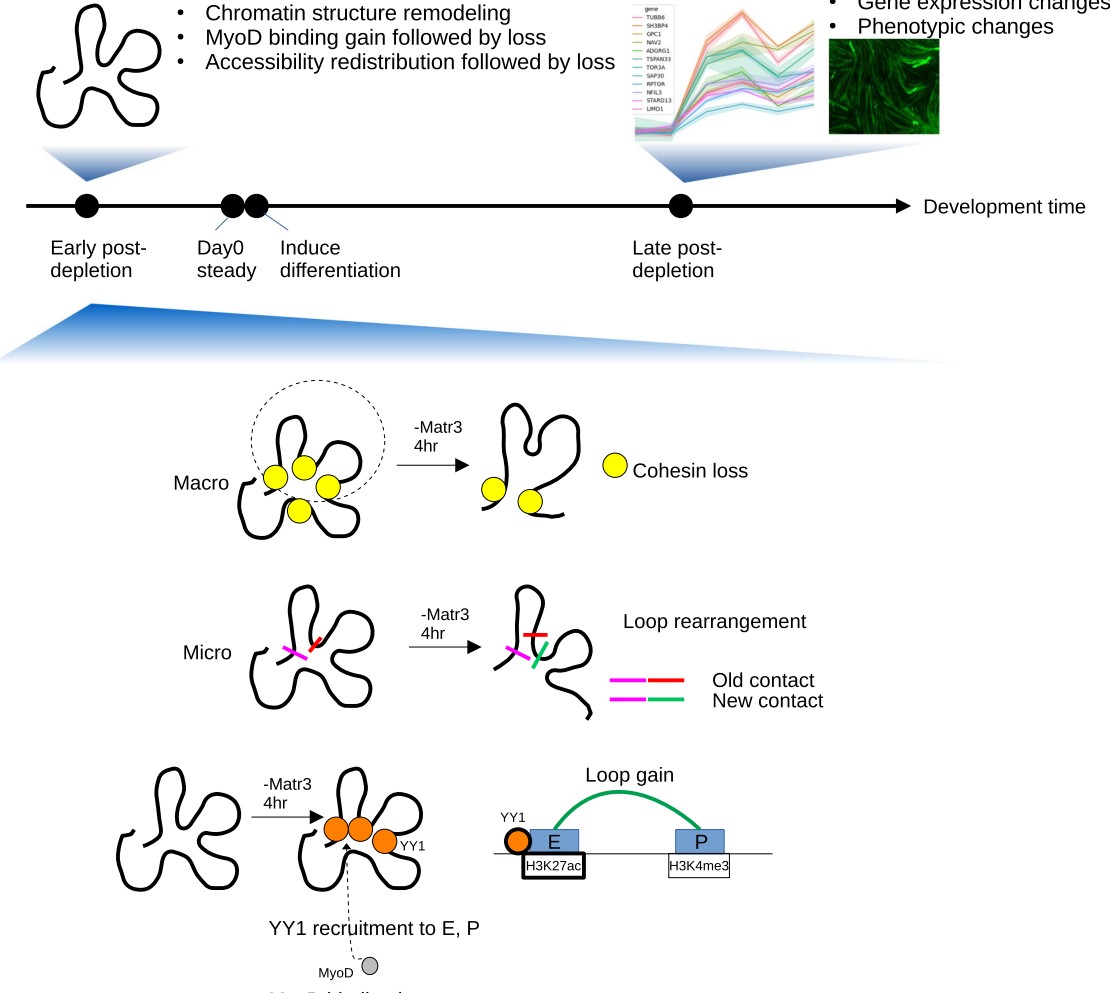

**Fig. 8 | Matr3 mediates differentiation through stabilizing chromatin accessibility and chromatin loop-domain interactions, and YY1 mediated enhancer-promoter interactions.** Acute depletion of Matr3 loss immediately reduced cohesin loading, and destabilized chromatin structural loops and affected long distance interactions. As a mechanism for partial compensation of loop re-

arrangement, YY1 was recruited to sites of cohesin loss, as well as enhancers and promoter sites, and formed new E-P loops, thereby disrupting the E-P loop landscape. During myogenesis, MyoD was recruited to those chromatin sites, leading to gene expression changes as skeletal muscle development progressed.

### Generation Matr3 KO bulk using Cas9/RNP transfection

RNP electroporation with 4D-Nucleofector™ were performed according to 4D-Nucleofector™ Protocol C2C12 cells (Lonza). Freshly thawed C2C12 cells were recovered and subculture 2 days before Nucleofection. Cells were harvested by trypsinization (0.25% trypsin, 37 °C) and

neutralization with culture medium. Next, cells were washed once with PBS and separated to $2 \times 10^5$/sample. Cas9/sgRNA RNP was generated by mixing 120 pM guide RNA (20-O-methyl analog and 30 phosphorothioate internucleotide modified-sgRNA synthetic from Synthego, UGAACUGAGUCGCUAUCCAG) with 61 pM Cas9-Alt-R protein (IDT),

and incubated at room temperature for 15 min. Meanwhile, $2 \times 10^5$ cells/sample were collected by centrifugation at 90 x g for 10 min, and resuspended carefully in 20 µl 4D-Nucleofector™ Solution (SE Cell Line 4D-Nucleofector™ X Kit, Lonza). Thereafter, RNP mix was added to the cells, and nucleofection was carried out in a 4D-nucleofector X Unit with program CD-137. Immediately after nucleofection, cells were supplemented with 80 µl prewarmed culture medium. Cells were then split to two wells of 6-well plate filled with 1.5 ml culture medium. Knockout efficiency using PCR and western blots were performed 2 days post RNP delivery. The gRNA sequences and the primers used for genotyping the clones are listed in Supplementary Table of primers and antibodies.

### Generation knock-in N-terminal dTAG-Matr3

**Plasmid design.** N-terminal fusions of FKBP[f36v] to endogenous matr3 were generated with double-strand break using Cas9/sgRNA followed by delivery of HDR donor cassette via crude-rAAV preparations as described elsewhere[17]. At the N-terminus, eGFP-p2a-FKBPf36v was inserted after the ATG start codon. In instances where the cut site of a gRNA was distal to the intended insertion site, the wobble bases of all amino acid sequences were re-coded to prevent unintended repair. Homology arms were designed to be ~900 bp for the N insertions.

### Generation of crude rAAV preps

Crude rAAV viral preps were generated as described[17]. Briefly, HDR templates consisting of eGFP-P2A-FKBP[f36v] flanked by homology arms were chemically synthesized ordered and cloned between the two Inverted Terminal Repeat (ITR) sequences of a pAAV2 vector to generate a transfer vector. Thereafter, the triple-transfection method was used to generate crude rAAV lysates[46].

### Generation dTAG knock-in C2C12 clones

C2C12 cells were electroporated with Cas9/sgRNA complexes targeting the HDR insertion site (with sgRNA protospacer sequence spanning the insertion site).

Electroporation was performed using the same setting as the Cas9/RNP transfection to generate Matr3 KO bulk, following the protocol of 4D-Nucleofector™ Solution (SE Cell Line 4D-Nucleofector™ X Kit, Lonza). In this setup, Cas9/sgRNA Ribonucleoprotein (RNP) mixture was generated with 120 pM guide RNA (2'-O-methyl analog and 3' phosphorothioate internucleotide modified-sgRNA, custom ordered from Synthego Corp., 5' to 3' sequence: UCUCUCGGUAGGGAUU-CACA) with 61 pM Cas9-Alt-R protein (IDT). Immediately after RNP electroporation, 150 µl crude rAAV viral prep (see above) was added to the cells (~10% of total culture medium), and the cells were returned to the 37 °C incubator for 3 days before expansion.

One week after electroporation/ AAV transduction, cells were sorted and top 0.2% GFP of parent cells were collected to generate the dTAG-Matr3 knock-in bulk and screen for single clones. To isolate single-cell clones, ~30 cells were seeded per 96-well plate. Single clones were expanded and genotyped for the desired knock-in by PCR on genomic DNA extracted with Quickextract solution (Lucigen). Clones showing biallelic knock-ins were expanded and frozen in LN2 at early passages. The gRNA sequences and the primers used for genotyping the clones are listed in Supplementary Table of primers and antibodies. Clones and bulk were validated by Western blot and Sanger sequencing.

### Western blot

Western blots were performed as described in previous paper with modification[13,47]. Protein was isolated from cells or myotubes using RIPA buffer (Boston Bioproduct BP115 containing 50 mM Tris, 150 mM NaCl, 10% glycerol, 1% NP40, 0.5% sodium deoxycholate, 0.1% SDS) with 0.5 mM PMSF (phenylmethylsulfonyl fluoride), 1 mM Na₃VO₄, and 1 mM NaF and complete EDTA-free protease inhibitor mixture (Roche Applied Science). Cell lysates were disrupted mechanically by passing

them through 25 G 5/8 needles 10 times and then centrifuged at 10,000 g for 5 min. The supernatant was collected, and protein concentrations were determined with BCA protein Assay Reagents (ThermoFisher Scientific, 23225) followed by measurement with NanoDrop. Protein samples were mixed with 4X Laemmli sample buffer (BIO-RAD # 161-0747) and same amount of protein (10 µg–40µg) was loaded and separated by Mini-PROTEAN® TGX™Precast Gels (4%–20%), Blots were probed with primary antibody and the Horseradish peroxidase conjugated secondary antibodies (antibodies used in western blots could be found in Supplementary table of primers and antibodies). Immunoblots were visualized using ECL plus reagent (GE Life Sciences), and the Fiji program was used to quantify protein abundance.

### Immunohistochemistry

For immunofluorescence staining with C2C12 myoblast and myotubes, 4-well Permanox Chamber Slides (Nunc™ Lab-Tek™ Chamber Slide System 177437, Thermo Fisher) were used for Olympus confocal imaging, and Greiner CELLSTAR® 96 well plates (Greiner 655090, Sigma) were used for Yokogawa CV7000 microscope. Cells were seeded on the plates/slides that were pre-treated with 0.3% gelatin, and kept in incubator for proliferation and differentiation. Cells were rinsed with PBS three times and then fixed with 4% PFA in PBS for 15 min at room temperature. After fixation, cells were rinsed twice with PBS before permeabilization with 0.25% Triton X-100 in PBS for 10 min. Next, cells were blocked with freshly made 5% normal donkey serum (or 5% goat serum) in PBST (PBS with 0.25% Triton X-100) for 1 h at room temperature, and were incubated with primary antibody in blocking solution overnight at 4 °C. The next day, after three washes with PBS, cells were incubated with secondary antibody in PBS for 1 h in dark at room temperature. After three washes with PBS, the cells were mounted in VECTASHIELD® Antifade Mounting Medium with DAPI (H-1200-10).

### Imaging and image analysis

**Olympus FV1000 confocal microscope imaging and analysis.** To get higher resolution of cell nucleus, cells were seeded on 4-well Permanox Chamber Slides (Nunc™ Lab-Tek™ Chamber Slide System 177437) and imaged with an Olympus FV1000 confocal microscope as described previously[47]. Confocal stacks were imaged with Olympus FV1000 equipped with 20×/0.85 N.A. and 100× 1.40 N.A. oil-immersion objectives. For double-labeling experiments, sequential scans of argon ion 488-nm and HeNe (633 nm for AlexaFluor 647) lasers were used to avoid bleed through between channels. The Fluoview "Hi-Lo" look-up table was used to set the maximal signal below saturation and set the background to near zero using the high voltage and offset controls. Z-series images were obtained at a 2-µm step size, and Kalman averaging was not used. Original images were saved as 12-bit OIB format and processed using FV1000 confocal software to generate maximum intensity projections (Z-projections). Images were adjusted for brightness and contrast using ImageJ/Fiji software. For each genotype, images were acquired using the same settings (power, gain, offset) at the same time.

### Yokogawa CV7000 microscope imaging and analysis

For high-throughput analysis, imaging was performed on a Yokogawa CV7000 microscope with 10x imaging using appropriate filter and laser settings (DCAF8/DMD - AF488: 488 excitations with 510/20 BP filter, MHC - AF647: 640 excitations with 661/20 BP filter). Images were acquired with an exposure of 1000 ms for all channels and captured with $2 \times 2$ binning at ±20 µm from the focal point with a step size of 10 µm (determined imaging parameters required to capture the whole depth of the muscle fiber). These images were automatically summed stacks in the CV7000 software and exported for analysis in Fiji, and subsequently data interpretation in Python (v3.85).

In order to create a per-channel light path correction image, multiple empty wells were stained and treated with anti-fade and were median projected in Image J. Then, the per-channel light path correction image was subtracted from the respective channel in all experimental wells using the ImageJ "Image Calculator" function. To gather muscle fiber ROIs, the MHC channel image was top hat filtered and a manual threshold was applied consistent across experiments. This muscle fiber mask was then used to calculate average MHC fiber intensity and applied to look at the DMD/DCAF8 staining intensity, since this staining should exist within only MHC+ fibers. The average per-channel fluorescence intensity was then recorded per well and imported to Python for analysis. Data was loaded into a pandas (version 1.1.3) dataframe and prepared for plotting. All plots were generated using the Seaborn (version 0.11.0) and matplotlib (version 3.3.2). Analysis of statistical significance was performed using scipy (version 1.7.3) Welch's $t$ test and Mann-Whitney U test with a $p$ value of 0.05 and assumption that KO fluorescence of the Target was less than the WT fluorescence of the Target, where the target is either DMD or DCAF8.

### Differentiation and fusion indices

Differentiation index and fusion index during C2C12 differentiation were quantified following the established protocol[14]. Nuclei were identified in ImageJ (version 1.45 f) using the StarDist algorithm[48], which robustly identified nuclei with defined borders. Applying this algorithm, nuclear area, and percent overlap with the aforementioned MHC mask were calculated. Nuclei with greater than a 78% percent overlap with MHC regions were assigned as within the muscle fiber, an assumption that was validated by manual counting. Data was loaded into a pandas (version 1.1.3) dataframe and prepared for plotting. All plots were generated using the Seaborn (version 0.11.0) and matplotlib (version 3.3.2). The differentiation index shown in Supplementary Fig. 3a was calculated as the percentage of nuclei contained within all myosin heavy chain (MHC) positive cells compared with the number of total nuclei within each image acquired by Yokogawa CV7000 microscope (above). At least 6 independent replicate wells were quantified for each genotype ($n = 6$). The fusion index shown in Supplementary Fig. 3b was calculated as the fraction of nuclei contained within MHC+ myotubes which had three or more nuclei, as compared to the number of total nuclei within each image. At least 6 independent replicate wells were quantified for each genotype.

### RNA-seq

Total RNA was isolated from cells using the RNeasy Plus Mini Kit (Qiagen). RNA sequencing libraries were prepared using Roche Kapa mRNA HyperPrep sample preparation kits from 100 ng of purified total RNA according to the manufacturer's protocol. The finished dsDNA libraries were quantified by Qubit fluorometer, Agilent TapeStation 2200, and RT-qPCR using the Kapa Biosystems library quantification kit according to manufacturer's protocols. Uniquely indexed libraries were pooled in equimolar ratios and sequenced on Illumina NovaSeq with paired-end 150 bp reads at the Dana-Farber Cancer Institute Molecular Biology Core Facilities.

### SLAM-seq

Thiol (SH)-linked alkylation for the metabolic sequencing of RNA (SLAM-seq) was performed as described[18,49]. Briefly, C2C12 cells were seeded the day before the labeling experiment that allows exponential growth (50%–80% confluency for experiment). $0.5 \times 10^6$ C2C12 cells per replicate was incubated with 500 nM dTAG47/DMSO for 4 h. (For washout experiments, after 4h. treatment with dTAG/DMSO, cells were washed twice with 1X PBS and grown for 5 days before S4U labeling). S4U labeling was performed by adding S4U to a final concentration of 100 µM for an additional hour. After labeling, cells were washed twice with 1X PBS, and lysed directly in TRIzol. Total RNA was extracted using Quick-RNA MiniPrep (Zymo Research) according to the manufacturer's instructions except including 0.1 mM DTT to all buffers. Thiol modification was performed by 10 mM iodoacetamide treatment followed by quenching with 20 mM DTT. RNA was purified by ethanol precipitation, and RNA-seq was performed as described above.

### ATAC-seq

ATAC-seq was performed as described[19]. Total 50,000 cells were washed with cold PBS and lysed using cold lysis buffer (10 mM Tris-HCl, pH 7.4, 10 mM NaCl, 3 mM MgCl2 and 0.1% IGEPAL CA-630). The pellet was then resuspended in the transposition reaction mix (25 µL 2× TD buffer, 2.5 µLTn5 Transposes (Illumina) and 22.5 µL nuclease-free water) and incubated at 37 °C for 30 min. Immediately following transposition incubation, DNA was purified using a Qiagen MinElute Kit. Transposed DNA fragments were amplified by PCR and libraries were sequenced on

on Illumina NextSeq500 with pair-end reads (2X 42 bp, 8 bp index) at the Dana-Farber Cancer Institute Molecular Biology Core Facilities.

### ChIP-seq

ChIP-seq was performed as described with modifications[10]. Briefly, ~1 × 10^7 cells per IP were crosslinked with 1% formaldehyde for 10 min at room temperature and followed by adding Glycine (2.5 M in dH2O) to final 0.125 M for 5 min. Cells were washed twice with ice-cold PBS, and then scraped off the plates into 15 ml Falcon tube. Nuclei were prepared using truChIP Chromatin Shearing Reagent Kit (Covaris). Chromatin was sonicated to around 200–500 bp in shearing buffer (10 mM Tris-HCl pH 7.6, 1 mM EDTA, 0.1% SDS) using a Covaris E220 sonicator. The sheared chromatin was diluted and adjusted to 150 mM NaCl and 1% Triton X-100, and incubated with antibody at 4 °C overnight. Protein A or G Dynabeads (Invitrogen) were added to the ChIP reactions and incubated for 3 h at 4 °C. Subsequently, Dynabeads were washed twice with low salt wash buffer(10 mM Tris-HCl PH7.4, 150 mM NaCl, 1 mM EDTA, 1% TritonX-100, 0.1% SDS, 0.1% sodium deoxycholate), twice with high salt wash buffer (10 mM Tris-HCl PH7.4, 300 mM NaCl, 1 mM EDTA, 1% TritonX-100, 0.1% SDS, 0.1% sodium deoxycholate), twice with LiCl buffer (10 mM Tris-HCl PH8, 1 mM EDTA,0.50% sodium deoxycholate, 0.5%NP-40, 250 mM LiCl) and twice with TE buffer(10 mM Tris-HCl pH 8.0, 1 mM EDTA, pH 8.0). The chromatin was eluted in SDS elution buffer (1% SDS, 10 mM EDTA, 50 mM Tris-HCl, pH 8.0) followed by reverse crosslinking at 65 °C overnight. ChIP DNA were treated with RNaseA and protease K, and purified using Phenol-chloroform extraction. 2–10 ng of purified ChIP DNA was used to prepare sequencing libraries, using NEBNext Ultra DNA Library Prep Kit for Illumina (NEB) according to the manufacturer's instructions. The finished ChIP-seq libraries were quantified by Qubit fluorometer, Agilent TapeStation 2200, and RT-qPCR using the Kapa Biosystems library quantification kit according to manufacturer's protocols. Uniquely indexed libraries were pooled in equimolar ratios and sequenced on Illumina NextSeq500 with single-end 75 bp reads at the Dana-Farber Cancer Institute Molecular Biology Core Facilities.

### CUT&RUN

CUT&RUN experiments were carried out as described[20,50] with modifications. Briefly, ~0.5 × 10^6 cells were collected and wash 3 times with Wash buffer (20 mM HEPES-KOH, pH 7.5,150 mM NaCl, 0.5 mM Spermidine, and Roche Complete Protease Inhibitor EDTA-free). Cells were captured with BioMagPlus Concanavalin A (Polysciences) for 10 min at room temperature and incubated with primary antibody (final concentration 1:100) on a nutator at 4 °C overnight. After incubation, samples were washed 3 times with Dig-wash buffer (0.025% digitonin, 20 mM HEPES-KOH, pH 7.5,150 mM NaCl, 0.5 mM Spermidine, and Roche Complete Protease Inhibitor EDTA-free). (Note: for primary antibodies from mouse, secondary antibody rabbit anti-mouse (final

1:100) was added at this step. Reactions were placed on the nutator at 4 °C for 1 h, and then washed 3 times with dig-wash buffer). Next, protein A-MNase was added at 700 ng/ml in Dig-wash buffer and incubated on a nutator at 4 °C for 1 h. Samples were washed again and placed in a 0 °C metal block (heating block sitting in wet ice). To activate protein A-MNase, $CaCl_2$ was added to a final concentration of 2 mM. The reaction was incubated at 0 °C for 30 min and stopped by addition of equal volume of 2XSTOP buffer (340 mM NaCl, 20 mM EDTA, 4 mM EGTA, 100 µg/ml RNase A and 50 µg/ml glycogen). The CUT&RUN fragments were released from the insoluble nuclear chromatin at 37 °C for 30 min. Digested chromatin in the supernatant were collected and then digested by proteinase K at 50 °C for 1 h. DNA was extracted by phenol chloroform extraction, ethanol precipitation.

## Library Preparation and Sequencing for CUT&RUN

For transcription factors (TF), libraries were constructed using the library preparation manual of "NEBNext® Ultra™ II DNA Library Prep Kit for Illumina", NEB E7645, with modification[51]. Briefly, dA-tailing temperature was decreased to 50 °C to avoid DNA melting, and the reaction time was increased to 1 h to compensate for lower enzymatic activity. After adaptor ligation, 1.87x volume of AMPure XP beads was added to the reaction to ensure high recovery efficiency of short fragments. After 12 cycles of PCR amplification (30 s@98 °C, 12 cycles of 10 s@98 °C, 10 s@65 °C, final extension 5 min @65 °C), the reaction was cleaned up with 1.2x volume of AMPure XP beads. For histone markers, libraries were prepared using NEBNext® Ultra™ II DNA Library Prep Kit for Illumina", NEB E7645, per manufacturer's instruction with modification. Briefly, after end repair and adapter ligation, 1.1x volume of AMPure XP beads was added to the reaction for DNA cleanup. After 12 cycles of PCR amplification (30 s@98 °C, 12 cycles of 10 s@98 °C, 10 s@60 °C, final extension 1 min @72 °C), the reaction was cleaned up with 1.1x volume of AMPure XP beads.

16–24 barcoded libraries were quantified and mixed at equal molar ratio. Library was loaded to a NextSeq 500/550 High Output Kit v2 (75 cycles), and sequenced in the NextSeq 500 platform. To enable determination of fragment length, paired-end sequencing was performed (2 × 42 bp, 6 bp index)

## Hi-C library preparation

Hi-C was performed as described with modification[52,53]. 5 million cells were crosslinked with 1% formaldehyde for 10 min and then quenched with glycine. Cells were lysed and then digested with DpnII overnight at 37 °C. Sticky ends were filled with dNTPs containing biotin-14-dATPs at 23 °C for 4 h. Furthermore, blunt ends were ligated using T4 DNA ligase at 16 °C for 4 h. Ligation products were treated with proteinase K at 65 °C overnight to reverse cross-linking and then purified using 2 consecutive phenol-chloroform extraction. Ligation products were confirmed by agarose gel.

Biotins were removed from un-ligated ends and then fragmented to average size of 200–400 bp by sonication. Fragmented DNAs were size-selected up to 400 bp using AMPure XP beads. At this step, Hi-C libraries were prepared using adapted NEBNext protocol to work after beads pulldown[53]. Biotin-tagged ends were pulled down using Dynabeads MyOne Streptavidin C1 (Thermo Fisher Scientific 65001). Standard Illumina library preparation protocol including end repair, A-tailing, and adaptor ligation was performed on beads with the NEBnext Ultra II kit (New England Biolabs E7645). An optimal PCR cycle for final library amplification using NEBnext Ultra II Q5 was determined, and between six and nine PCR cycles were used in our study. Amplified PCR libraries were purified with 1.5X AMpure XP beads and were quantified by a Qubit fluorometer and Qubit dsDNA HS kit (Thermo Fisher Q32851). For quality control, a small aliquot of the final Hi-C libraries (1.5 µL) was digested with ClaI enzyme at 37 °C for one hour and quantified approximate% uncut DNA (correlates to the fraction of non-ligation products) from Bioanalyzer analysis. Illumina

NovaSeq 50-bp paired-end sequencing (PE50) was used to obtain ~400 million reads for each replicate.

## Data processing and analysis

**Gene expression analysis from RNA-seq and SLAM-seq.** RNA-seq experiments were processed by the HISAT pipeline, which includes alignment, filtering, and gene quantification steps. Differential expression analysis was carried out using DESeq2 using default settings[54]. We performed SLAM-seq at 4 h. post dTAG47 exposure and quantified nascent RNA transcription using the SlamDunk pipeline[49,55]. Differential analysis of SLAM-seq data was performed on the transcript read counts with thymine-to-cytosine (T > C) conversions following s4U metabolic labeling (newly synthesized RNA) and also on the total transcript read counts (total RNA). DESeq2 was again used for differential nascent transcript analysis.

## CUT&RUN data normalization and processing

We used CUT&RUNTools[21] to process MyoD, YY1, CTCF, and Rad21 CUT&RUN experiments from raw fastq reads. This process included reads trimming, alignment to the reference genome (mm10), BAM file duplicate marking, and fragment filtering based on fragment size. Next, we performed background-based data normalization on the group of experiments to be normalized together (i.e., Matr3 dTAG47 & DMSO, or Matr3 WT & KO of a given antibody). For experiments per group, we pooled all experiments' BAM files and performed MACS2[56] peak calling to get N peaks. Peak-flanking regions were obtained, and stored for the purpose of normalization. Bamliquidator was called to compute the number of reads in each peak-flanking region in each experiment. DESeq2's estimateSizeFactor()[54] was called on the matrix of peak-flanking region reads over all experiments to calculate a scale factor per experiment. With the scale factor computed, the original BAM file was subsampled at a rate equal to the scale factor to get a normalized BAM file. Finally, MACS2 was called to generate peaks from the normalized BAM file. A normalized bigwig file was used for visualization.

## ATAC-seq data normalization and processing

ATAC-seq data were processed using a procedure similar to CUT&RUN described above, which included reads trimming, alignment to the reference genome, BAM duplicate marking, and fragment size filtering. A special note is that we kept paired-end fragments that are <150 bp instead of <120 bp setting for TF CUT&RUN. For ATAC-seq data normalization, we adopted the same background-based data normalization as used in CUT&RUN (see CUT&RUN section).

## Differential binding analysis of CUT&RUN and ATAC-seq data

To identify differential binding between dTAG47 & DMSO or WT & KO, experiments to be compared were first normalized using background-based data normalization (see above). Next, we pooled the BAM files and called peaks using MACS2 to generate a set of common regions on which to test differential binding. Bamliquidator was used to compute the number of reads in each common region in each experiment. Finally, DESeq2 was called to identify regions that exhibit differential binding.

## ChIP-seq analysis

For analysis of CTCF and Rad21 ChIP-seq, we used a ChIP-seq data processing pipeline to perform reads trimming, alignment to the reference genome (mm10), and BAM file duplicate marking. Next, we performed background-based data normalization on the group of experiments to be normalized together (i.e., Matr3 dTAG47 & DMSO, or Matr3 WT & KO of a given antibody), similar to the normalization procedure used in CUT&RUN data. Once samples were normalized, we proceeded to call peaks using MACS2, and then differential binding analysis using DESeq2, same as described in the CUT&RUN paragraph.

## Gene set enrichment analysis

We mapped differential ATAC-seq peaks to the nearest genes that are 25 kb from transcription start site (TSS). Each differential peak contributed a score to the gene that equals to the $1/n * (-10\log P)$ where n is the number of genes within 25 kb vicinity of the peak, and P is the *P* value significance of the differential peak. We next input the top scoring genes to DAVID Functional Annotation Tool (https://david.ncifcrf.gov/tools.jsp) for gene enrichment analysis using default settings. To ensure the robustness of the gene enrichment results, we also performed enrichment using SEEK (https://seek.princeton.edu). Here the top scoring genes were input as a query and co-expressed genes to the query were identified and were used to perform enrichment analysis. We used the default settings (top 500 co-expressed genes and GO: biological process).

## Hi-C processing and analysis

We used HiC-Pro[57] to align reads to mm10 reference genome assembly. By providing the DpnII restriction enzyme digestion sites and upon further filtering, HiC-Pro returns valid interaction pairs for each of four conditions: dTAG47, DMSO, WT, and KO. Next, we used the HIFI tool[58] to impute interactions for our Hi-C data based on the observed over expected interaction frequency matrix. The interaction frequency matrix was at restriction fragment resolution, roughly equivalent to ~5 kb resolution between contact sites. HIFI employs a Markov random field-based imputation to fill in missing values, and smooth neighboring values in the contact matrix to enhance loop detection. Initial evaluation found that this additional imputation step improved the detection of loops from Hi-C data without loss of resolution and increased the agreement between different loop calling methods (Supplementary Fig. 12). We next performed loop calling on the imputed interaction frequency matrix using the Hiccup similar procedure that is implemented within HIFI.

For differential loop analysis, we had three Hi-C replicates per DMSO and dTAG47 groups. After calling loops in individual replicate samples, we pooled loops to get a union of interaction regions on which to do differential analysis. For each $(i,j)$ interaction pair, we went back to each sample to compute the Hiccup[26] score in each sample which was defined by $score(i,j) = P(i,j)/ \max(D(i,j), H(i,j), V(i,j), BL(i,j))$, where $P(i,j)$ is the average interaction frequency (IF) in the peak. $D(i,j)$, $H(i,j)$, $V(i,j)$, and $BL(i,j)$ are several types of flanking regions to which $P(i,j)$ is compared against.

Hiccup scores were z-score normalized within each sample across all pairs.

Differential loop scores were obtained by: $z_{DMSO1}(i,j) - z_{dTAG1}(i,j), z_{DMSO2}(i,j) - z_{dTAG2}(i,j), z_{DMSO3}(i,j) - z_{dTAG3}(i,j), \ldots$, for all combinations of DMSO and dTAG replicates, resulting in a set of difference scores $DS$. A final meta-differential loop score $Z(i,j)$ was returned by Stouffer z-score meta-analysis method: $Z(i,j) = \frac{1}{\sqrt{k}} \sum_k z_{DMSOx}(i,j) - z_{dTAGx}(i,j)$ where $k = |DS|$. $Z(i,j)$ that correspond to $P < 0.05$ are deemed as significant.

For compartment analysis, A/B compartments were identified using the "runHiCpca.pl" script in HOMER[59] at the resolution of 50 kb. The "±" sign of the compartment was determined by TSS enrichments, where compartments showing enrichment of TSS were assigned the '+' sign and those showing depletion of TSS were assigned the '−' sign. To visualize the changes in compartmentalization strength, we generated and compared the saddle plots. First, for each chromosome, we removed the 1% genomic bins with the lowest sequencing coverage in order to remove the bias caused by insufficient coverage. Then we ranked the remaining genomic bins by the PC1 scores from high to low. We reordered the rows and columns of the contact matrix according to the same ordering. Then the contact map was coarse-grained into a 100*100 matrix, where the element (m, n) represents the mean interaction frequency between bins of the m-th percentile and the n-th percentile. The average of the coarse-

grained contact matrices from all chromosomes were then plotted as the saddle plot. The obs/exp contact matrices at 50 kb were used for this analysis. To quantify the intra- and inter-compartment interaction strength, the average interaction strength for genomic bins with PC1 values in top 25% and bottom 25% are used to measure the AA (top left), BB (bottom right) and AB (top right) interaction strengths. To visualize the differences in saddle plots of two samples, we plotted the fold change of the two contact matrices in an element-wise way after log2 transformation

## Visualization of CUT&RUN heatmaps and Hi-C interactions

To generate comparative heatmaps between CUT&RUN groups (DMSO vs dTAG47), we used Deeptools's computeMatrix and plotHeatmap functions on the list of differential binding sites. For visualizing Hi-C interaction matrix, we used the HIFI visualization capability to generate a fragment-resolution interaction heatmap. To interactively navigate the Hi-C data, we used the Washington University Epigenome Browser (http://epigenomegateway.wustl.edu/). We prepared the interaction data in the tabix and longrange format as required by the browser, and loaded the tracks. We chose the "arc" representation to visualize locus-specific interactions. The "heatmap" mode was chosen to visualize all interactions in a given region, omitting interactions that are outside, and visualizing gained and lost interactions in shades of greens and reds respectively.

## Reporting summary

Further information on research design is available in the Nature Portfolio Reporting Summary linked to this article.

## Data availability

The data that support this study are available from the corresponding author upon reasonable request. Hi-C, CUT&RUN, ChIP-seq, ATAC-seq, RNA-seq and SLAM-seq data sets generated this study have been deposited in the GEO database, under accession code GSE247105. Source data are provided with this paper.

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

## Acknowledgements

We thank the Molecular Biology Core Facility at DFCI, HESC Core Facility, BCH/PCMM Microscopy Core, and Viral core at BCH. We thank Drs. Nan Liu for CUT&RUN reagents and discussion, Manuela Gussoni for myotube immunostaining antibodies and discussion, Louis M. Kunkel Lab for DMD antibody, Stephen Tapscott Lab for MyoD antibody, Kevin P. Campbell and Mary E. Anderson for DMD complex antibodies and ana-lysis, and Jun Qi for dTAG47. Computational analyses were conducted on the O2 High Performance Compute Cluster, supported by the Research Computing Group at Harvard Medical School. S.H.O. is an Investigator of the HHMI.

## Author contributions

T.L., Q.Z. and S.H.O. designed the study. T.L. performed the experiments, and analyzed and interpreted the data. Q.Z. performed computational analyses and interpreted the results Y.K. analyzed Hi-C compartments. G.Y. helped with computational analyses tools. T.B. performed Yoko-gawa CV7000 imaging analysis. T.M.S. helped Yokogawa CV7000 imaging analysis tools. S. W. helped performed computational analysis. H.C. helped generated the plasmid for Matr3 knockout clones. S.M. helped design the plasmids for knock-in dTAG-Matr3.

## Competing interests

The authors have no competing interests.
