## [Peer Review File · Nature Communications]

Matrin3 mediates differentiation through stabilizing chromatin loop-domain interactions and YY1 mediated enhancer-promoter interactionsREVIEWER COMMENTS

Reviewer #1 (Remarks to the Author):

This study explores the role of Matr3 in muscle development and its impact on chromatin structure. Traditionally, Matr3 has been considered to be involved in RNA splicing and transport. However, this research demonstrates that Matr3 regulates chromatin structure during the differentiation processes, through interactions with architectural proteins cohesin/Rad21. Using Matr3 knockdown technology, the experiments show that Matr3 plays a critical role in the muscle differentiation of the C2C12 myoblast. Loss of Matr3 results in early dynamic changes in chromatin accessibility and MyoD occupancy. Furthermore, after Matr3 knockdown, significant chromatin loop rearrangements were observed, along with changes in YY1 occupancy sites, which are related to differences in chromatin loop interactions. The study also found that the loss of Matr3 affects the formation of YY1-mediated enhancer-promoter loops. These findings suggest that Matr3 depletion may impact chromatin loop formation and structure by affecting cohesin and YY1 occupancy. Additionally, the loss of Matr3 is closely related to the expression and differentiation of muscle development-associated genes.

In summary, this research reveals the critical role of Matr3 in the context of muscle cell differentiation, directly linking changes in proteins within nuclear compartments to alterations in chromosomal architecture. By influencing chromatin accessibility and the occupancy of cohesin and YY1, Matr3 impacts chromatin structure to orchestrate cell differentiation. The study appears to be well-designed and provides valuable insights into the role of Matr3 in muscle development and chromatin structure. However, the conceptual novelty of this study is somewhat diminished as the authors have published a similar story regarding erythroid precursor differentiation. Future research could investigate how Matr3 stabilizes chromatin during cell differentiation, which could potentially enhance the quality of this study.

Major:

1. The authors should investigate the specific molecular mechanisms behind Matr3's interaction with architectural proteins such as cohesin/Rad21 to better understand its role in chromatin loop formation and regulation.
2. For Figure 1, the authors should include ATAC-seq at differentiation day 0 and day 4 to demonstrate how Matr3 KO influences overall DNA accessibility during C2C12 myoblast differentiation. It is important to note that Figure 1C displays only a small fraction of nuclei within the myotubes that are undergoing true differentiation. The question arises as to how a bulk assay can ensure the capture of biological events in these myonuclei.
3. The authors should elucidate whether the observed effects result from a combination of both RNA processing/splicing and chromatin structure alterations?
4. The authors could perform rescue experiments by reintroducing Matr3 into Matr3 KO C2C12 cells to assess whether the observed chromatin and gene expression changes can be reversed. This would help confirm the specificity of Matr3's effects on chromatin structure and gene regulation.
5. Please revise the label: "Chromatin structure remodeling MyoD binding gain followed by loss Accessibility redistribution followed by loss". Please clarify whether the early increased MyoD binding (Figure 3b) occurs after YY1 recruitment, and if so, why it "followed by gradual loss of MyoD binding (Figure 3b)"?
6. Does Matr3 specifically assist in the transcription of certain genes (such as DMD, DCAF8)? What is the basis for this specificity?

Minor:

1. To help readers understand the differences in measurement values and the quantity of samples, change the bar plots in the figures to bar plots with dots.
2. Measurements of cell diameter and myotube fusion index should be applied to quantify cell differentiation.
3. The authors should indicate the cut-off value of log2FC in the volcano plots.
4. The GAPDH band in Fig 2c is not acceptable; please provide a better quality image.
5. Please relabel H3K27AC and H3K27me3 in Figure 7.
6. The manuscript should provide web links for raw/processed sequencing data and code.

Reviewer #2 (Remarks to the Author):

The manuscript entitled "Matrin3 mediates differentiation through stabilizing chromatin accessibility and chromatin loop-domain interactions, and YY1 mediated enhancer-promoter interactions", showed that the inner nuclear protein MATR3 is important for myogenesis and chromatin organization. The authors used C2C12 cell differentiation as a model, and investigated gene expression, chromatin accessibility and chromatin-chromatin interaction via multiple genomic assays. They found that acute MATR3 degradation leads to little change on nascent RNA expression, but markedly alterations of chromatin accessibility, transcription factors (including MyoD and YY1) binding, and chromatin loops. Overall, this study is comprehensive and some findings are interesting. However, the mechanisms on how MATR3 regulates differentiation and chromatin architecture remain unclear. Please see specific comments below.

Major

1. Why MATR3 loss leads to the changes of MyoD and YY1 binding? Are these sites bound by MATR3, directly or indirectly, and prevented the occupancy of transcription factors? ChIP-seq or Cut&RUN with MATR3 should be important for this.
2. It is interesting that the fast loss of MATR3 resulted into the increase of MyoD binding. MyoD is the master regulator of muscle cell differentiation. A study recently reported that MyoD may function as the 3D genome structure organizer for muscle cell identity (PMID: 35017543). But why MATR3 knockout leads to defects in myogenesis? This should be discussed.
3. The authors found that acute degradation of MATR3 led to little change in nascent RNA and elicited changes of expression at later developmental stages. However, there are many genes changes as measured by total RNA (Fig 2d: right), and this was not discussed in the manuscript.
4. How does MATR3 stabilize YY1-mediated promoter-enhancer interactions? As MATR3 and YY1 both bind RNAs, is it possible that MATR3 stabilizes these chromatin loops via some types of non-coding RNAs (e.g., lncRNA or eRNA)?
5. Thy rearranged loops were concentrated on chr 7, 14 and X? What unique genomic features with these three chromosomes? Or these chromosomes have critical genes in myogenesis? This is an interesting finding, but did not expand in current version.

Minor

1. Fig 1c : quantification is needed.
2. The conclusion "depletion of Matr3 leads to aberrant muscle differentiation" is overstated, as there is no in-vivo data to support "muscle differentiation".
3. Fig 3a-b: Scales should be added; to show the difference, average profiles (as those shown in Fig 4a) should be also presented. Fig 3c-d, gene symbols and heatmap cells are non-corresponding. Any statistical test was used for Fig3h-3i?
4. Fig 4a and 5c-5e: Scales should be added.
5. From the data presented in Fig S7, it remained unclear that "chromatin loops were correlated with differential open chromatin accessibility. What the colors mean in Fig S7a? There is no clear trend in this figure.
6. The quality of Fig 2c should be improved.

REVIEWER COMMENTS

Reviewer #1 (Remarks to the Author):

This study explores the role of Matr3 in muscle development and its impact on chromatin structure. Traditionally, Matr3 has been considered to be involved in RNA splicing and transport. However, this research demonstrates that Matr3 regulates chromatin structure during the differentiation processes, through interactions with architectural proteins cohesin/Rad21. Using Matr3 knockdown technology, the experiments show that Matr3 plays a critical role in the muscle differentiation of the C2C12 myoblast. Loss of Matr3 results in early dynamic changes in chromatin accessibility and MyoD occupancy. Furthermore, after Matr3 knockdown, significant chromatin loop rearrangements were observed, along with changes in YY1 occupancy sites, which are related to differences in chromatin loop interactions. The study also found that the loss of Matr3 affects the formation of YY1-mediated enhancer-promoter loops. These findings suggest that Matr3 depletion may impact chromatin loop formation and structure by affecting cohesin and YY1 occupancy. Additionally, the loss of Matr3 is closely related to the expression and differentiation of muscle development-associated genes.

In summary, this research reveals the critical role of Matr3 in the context of muscle cell differentiation, directly linking changes in proteins within nuclear compartments to alterations in chromosomal architecture. By influencing chromatin accessibility and the occupancy of cohesin and YY1, Matr3 impacts chromatin structure to orchestrate cell differentiation. The study appears to be well-designed and provides valuable insights into the role of Matr3 in muscle development and chromatin structure. However, the conceptual novelty of this study is somewhat diminished as the authors have published a similar story regarding erythroid precursor differentiation. Future research could investigate how Matr3 stabilizes chromatin during cell differentiation, which could potentially enhance the quality of this study.

Major:

1. The authors should investigate the specific molecular mechanisms behind Matr3's interaction with architectural proteins such as cohesin/Rad21 to better understand its role in chromatin loop formation and regulation.

To study molecular mechanisms, we mapped Matr3 binding profiles by CUT&RUN (Figure 7). Using Matr3 antibody recently demonstrated to permit chromatin localization (Wang B. et al., 2023, PMID: 37000624), we performed Matr3 CUT&RUN. We detected >20,000 genome-wide sites of Matr3 occupancy in the wild-type cells, of which >40% overlap with YY1 sites and 33% overlap with architectural CTCF binding sites. Upon Matr3 depletion, co-occupancy sites shared between Matr3 and Rad21, YY1, CTCF were directly affected, indicating functionally relevant relationships among these factors. Furthermore, Matr3 directly contributed to loop rearrangement. Genomic loci harboring loop rearrangements were heavily enriched for sites of Matr3 occupancy. Moreover, E-E and E-P loop anchors that exhibit Matr3 occupancy at anchors were more likely to be gained after Matr3 depletion, and harbor loop gain/loss imbalance characteristic of loop rewiring. Taken together, these findings suggest that the changes in occupancy of Rad21, YY1, CTCF, MyoD, and chromatin accessibility are the consequence of loss of Matr3 at these locations. The role in chromatin loop formation is inseparable from Matr3 occupancy genome-wide as well. We believe these new data

regarding Matr3 occupancy, as determined by CUT&RUN, provide a critical evidence regarding mechanism requested by this reviewer.

2. For Figure 1, the authors should include ATAC-seq at differentiation day 0 and day 4 to demonstrate how Matr3 KO influences overall DNA accessibility during C2C12 myoblast differentiation. It is important to note that Figure 1C displays only a small fraction of nuclei within the myotubes that are undergoing true differentiation. The question arises as to how a bulk assay can ensure the capture of biological events in these myonuclei.
For Figure 1, the authors should include ATAC-seq at differentiation day 0 and day 4 to demonstrate how Matr3 KO influences overall DNA accessibility during C2C12 myoblast differentiation.

Matr3 KO impacts cell differentiation by accelerating the process. Accordingly, chromatin accessibility for Matr3 KO at day 0 will resemble accessibility of WT cells at day 4 (See below). This role of Matr3 in accelerating myogenesis differentiation is consistent with what we previously reported in MEL cells and embryonic cells (Cha's paper Figure 7, Cha et al., 2021, PMID: 34716321). Therefore, we did not expand on this point in this manuscript. The focus of the current study is to probe how Matr3 loss alters chromatin architecture at the early times, in order to exclude secondary effects of cell development and focus specifically on primary effects of Matr3 depletion. Using the dTAG system to deplete Matr3 in a rapid fashion, we capture direct changes.

It is important to note that Figure 1C displays only a small fraction of nuclei within the myotubes that are undergoing true differentiation. The question arises as to how a bulk assay can ensure the capture of biological events in these myonuclei.

In C2C12 cells at Day 4 differentiation, the typical percentage of differentiated cells is ~ 50%, as reported by Quinn et al (Published Figure 4b, Nat Commun. 2017, PMID: 28569755) and confirmed in our experiments. To gather more images and increase sample size, we repeated immunostaining experiments and updated Figure 1c. Following the established protocol (Quinn et al. Nat Commun. 2017), we performed differentiation assays, which quantified fraction of nuclei contained within all MHC+ cells, as compared with the number of total nuclei (Supplemental Figure 3a), and the fusion assay, which quantified the fraction of nuclei contained within MHC + myotubes with >3 nuclei, as compared to the number of total nuclei (Supplemental Figure 3b). We also quantified myotube density in each condition (Figure 1c, right). We observed myotube density (the

myotube number in the same area across samples, and at least 6 replicates) significantly increased in Matr3 knockout. These findings support the conclusion that Matr3 KO leads to defects in myogenesis, that mimic the DMD hypertrophy phenotype.

3. The authors should elucidate whether the observed effects result from a combination of both RNA processing/splicing and chromatin structure alterations?

The principal contribution of the current manuscript involves an assessment of direct consequences of Matr3 depletion on chromatin architecture. How changes in chromatin might impact RNA processing/splicing, we believe, is an interested topic, but beyond the scope of this already comprehensive report.

Previously, we failed to observe a significant correlation between splicing events and gene expression upon perturbation of Matr3, revealing only a small subset of alternative splicing events that appeared to be associated with the transcriptome shift of Matr3 KO, compared with the significant association that found with the splicing regulator Ptp1 (Cha's paper figure S3b, Cha, et al., 2021, PMID: 34716321).

Further investigation of the interplay between splicing and chromatin structure will require in-depth experiments which are beyond the scope of this work.

4. The authors could perform rescue experiments by reintroducing Matr3 into Matr3 KO C2C12 cells to assess whether the observed chromatin and gene expression changes can be reversed. This would help confirm the specificity of Matr3's effects on chromatin structure and gene regulation.

To test whether restoration of Matr3 protein reverses gene expression changes, we performed washout experiments (Supplemental Figure 5). The level of Matr3 protein recovered after washout (Supplemental Figure 5a), and the DE gene expression level was rescued (Supplemental Figure 5b), indicating that gene expressions could be reversed. We also made an effort to test other markers after restoration of Matr3 protein. We initially tested Rad21, MyoD, YY1 chromatin binding profiles. After washout, differential binding of Rad21, MyoD, YY1 upon Matr3 depletion was largely reversed. We provide chromatin binding profiles for the reviewer (below) and have not included them in the revised manuscript.

5. Please revise the label: “Chromatin structure remodeling MyoD binding gain followed by loss Accessibility redistribution followed by loss”. Please clarify whether the early

increased MyoD binding (Figure 3b) occurs after YY1 recruitment, and if so, why it “followed by gradual loss of MyoD binding (Figure 3b)”?

Please revise the label: “Chromatin structure remodeling MyoD binding gain followed by loss Accessibility redistribution followed by loss”.

We revised the label in in Figure 8 (previous Figure7)

Please clarify whether the early increased MyoD binding (Figure 3b) occurs after YY1 recruitment

To address a link between increased YY1 binding and MyoD recruitment, we explored MyoD occupancy following Matr3 and YY1 knockout (Supplemental figure 10). We observed that upon YY1 loss, MyoD occupancy decreased, implying that early recruitment of MyoD is dependent on YY1.

if so, why it “followed by gradual loss of MyoD binding (Figure 3b)”?

We addressed this question in the discussion. We don't know the precise mechanism but may suggest several hypotheses. First, gradual loss of MyoD occupancy might be related to the overall decrease in chromatin accessibility accompanying differentiation. The “permissive fate” model (Martin et al., 2021, PMID: 33407811) describes that CREs of all lineage outcomes start out being in an accessible state, keeping these elements primed for subsequent activation. As cell differentiation proceeds, accessibility is restricted for alternative fates. Second, loss of MyoD occupancy may reflect the failure of skeletal muscle cells to regenerate. Loss of the ability to regenerate is a feature of muscle wasting syndromes (i.e. cachexia).

6. Does Matr3 specifically assist in the transcription of certain genes (such as DMD, DCAF8)? What is the basis for this specificity?

Matr3 CUT&RUN reveals genome-wide chromatin occupancy. The protein does not merely impact a few genes, but rather targets the EP loop landscape broadly by stabilizing interactions. Its pervasive role in recruiting MyoD and modulating YY1-mediated EP loops indicates that its impact is broad. Nonetheless, we note a preponderance of chromatin loop rearrangement events at specific chromosomal regions, namely chr7, chrX, and chr14, as also noted by Reviewer 2. These convey some skeletal muscle specific effects upon Matr3 depletion. In addition, we observed immediate chromatin loop changes concentrated on the chromosome X, where the DMD is located.

Minor:

1. To help readers understand the differences in measurement values and the quantity of samples, change the bar plots in the figures to bar plots with dots.

We have re-plotted the quantification in Western blots (Figure1), DMD and DCAF8 immunostaining quantification data (Figure1).

2. Measurements of cell diameter and myotube fusion index should be applied to quantify cell differentiation.

We have performed the assays to quantify Fusion index and differentiation index (Supplemental Figure 3).

3. The authors should indicate the cut-off value of log₂FC in the volcano plots.

Done, please see vertical blue lines in the volcano plots where log₂FC>0.4 is added.

4. The GAPDH band in Fig 2c is not acceptable; please provide a better-quality image.

We repeated the experiment in Figure 2c and provided new data.

5. Please relabel H3K27AC and H3K27me3 in Figure 7.

Done.

6. The manuscript should provide web links for raw/processed sequencing data and code.

We are now providing the web links for raw/processed sequencing data and code.

Reviewer #2 (Remarks to the Author):

The manuscript entitled “**Matrin3 mediates differentiation through stabilizing chromatin accessibility and chromatin loop-domain interactions, and YY1 mediated enhancer-promoter interactions**”, showed that the inner nuclear protein MATR3 is important for myogenesis and chromatin organization. The authors used C2C12 cell differentiation as a model, and investigated gene expression, chromatin accessibility and chromatin-chromatin interaction via multiple genomic assays. They found that acute MATR3 degradation leads to little change on nascent RNA expression, but markedly alterations of chromatin accessibility, transcription factors (including MyoD and YY1) binding, and chromatin loops. Overall, this study is comprehensive and some findings are interesting. However, the mechanisms on how MATR3 regulates differentiation and chromatin architecture remain unclear. Please see specific comments below.

Major

1. Why MATR3 loss leads to the changes of MyoD and YY1 binding? Are these sites bound by MATR3, directly or indirectly, and prevented the occupancy of transcription factors? ChIP-seq or Cut&RUN with MATR3 should be important for this.

We thank the reviewer for this insightful comment. As noted above for Reviewer 1, we mapped Matr3 occupancy by CUT&RUN (Figure 7). We found that the early changes detected in Rad21, YY1, MyoD, chromatin accessibility corresponded to regions occupied by Matr3, indicative of a direct relationship. In addition, we addressed the link between increased YY1 binding and MyoD recruitment (Supplemental figure 10). We found that when both YY1 and Matr3 are depleted, MyoD occupancy decreased, compared to Matr3-depleted cells but with YY1 present. This finding suggests that early recruitment of MyoD is dependent on YY1. Based on these data, we infer Matr3 loss destabilizes the cohesin-CTCF complex and reduces cohesin Rad21 occupancy. YY1 is recruited to the

chromatin sites, and the recruitment of YY1 contributes to recruitment of MyoD, all culminating in changes in gene expression.

2. It is interesting that the fast loss of MATR3 resulted into the increase of MyoD binding. MyoD is the master regulator of muscle cell differentiation. A study recently reported that MyoD may function as the 3D genome structure organizer for muscle cell identity (PMID: 35017543). But why MATR3 knockout leads to defects in myogenesis? This should be discussed.

We appreciate the comment from the reviewer, and we added this in the discussion. We believe that the short-term increase of MyoD occupancy reflects accelerated differentiation, a phenomenon we previously observed on Matr3 loss (Cha et al., 2021, PMID: 34716321), but Matr3 impacts many genes in the myogenic pathway. Therefore, the program of differentiation is perturbed.

3. The authors found that acute degradation of MATR3 led to little change in nascent RNA and elicited changes of expression at later developmental stages. However, there are many genes changes as measured by total RNA (Fig 2d: right), and this was not discussed in the manuscript.

Differences between nascent RNA and total RNA reflect the measurement window. Nascent RNA assay measures the newly synthesized RNA within the 1hr as well as RNA stability. In contrast, total RNA measures mRNA changes during 4hrs dTAG drug; therefore, it scores accumulated changes over a longer period. (Please see the details in SLAM-seq in “Method and Material”, and Gene expression analysis from RNA-seq and SLAM-seq, “Data processing analysis”). Therefore, it is reasonable that nascent RNA change is less than the total RNA change, but it should be more representative of the direct change in the target. Statistically, because nascent RNA detects fewer RNAs than the total RNA method, the number of significant genes will be less as a result. Despite these differences, we mention in text that when we looked further down the significant gene list, many of DE genes were similar between nascent and total RNA, thus providing confidence in the data. Further, if a filter of $\log_2FC > 0.4$ is applied, we observe that the number of detected DE genes is similar between nascent RNA vs. total RNA (10 and 9 genes respectively, out of over 20,000 genes).

4. How does MATR3 stabilize YY1-mediated promoter-enhancer interactions? As MATR3 and YY1 both bind RNAs, is it possible that MATR3 stabilizes these chromatin loops via some types of non-coding RNAs (e.g., lncRNA or eRNA)?

We appreciate this question. The CUT&RUN data clearly indicate that Matr3 is associated with chromatin. Our analysis reveals >40% of YY1 occupied sites are shared with Matr3 (Figure 7a). It is possible that RNA binding is involved and we cannot currently rule this out. Reviewer has raised an interesting possibility that Matr3 stabilizes these chromatin loops through non-coding RNAs. We believe that this is a plausible hypothesis. A recent paper (Zhang et al., 2023, PMID: 37381832) reported that MATR3 binds to antisense LINE1 (AS L1) RNAs, and formed a meshwork that gathers chromatin in the nucleus, and affects higher-order chromatin organization. A recent report also shows hnRNPM (a sister protein to MATR3) can bind LINE1 elements to regulate interferon response, reflecting a

common function (Zheng et al, 2023, PMID: 36865202). In our results, some accessible regions and MyoD targets affected by Matr3 loss are critical noncoding RNAs (see Malat1 in Figure 3c and Linc-MD1 in Figure 3d). Interestingly, the Mphosph8 locus, which exhibits changes in chromatin accessibility and TFs (highlighted in Figure 6), has been reported to be regulated by a LINE-1-retrotransposon. Mphosph8 is a member of Human silencing hub (HUSH) complex that transcriptionally represses L1 retrotransposons. HUSH and L1s may be part of a network that cooperates with Matr3 to regulate transcription.

5. Thy rearranged loops were concentrated on chr 7, 14 and X? What unique genomic features with these three chromosomes? Or these chromosomes have critical genes in myogenesis? This is an interesting finding, but did not expand in current version.

We have added the chromosomes that are enriched for rearranged loops to the discussion session. We collected genes located in rearrangement hotspots (Supplemental Figure 11) in chrs 14, 7, X, encompassing in total ~500 genes. We next performed GSEA analysis to identify associated GO concepts. On top of the enriched list, Cytoband Chr19q13 is the human syntenic segment for chr7 in mouse, suggesting this is a conserved segment, covering TGFB1, LTBP4, and has both cancer-related and muscle disease-related genes (DMPK, DMWD, SIX5 related to Myotonic Dystrophy Type 1). The X chromosome regions are almost completely syntenic in both species, and are enriched in the disease concept X-linked Recessive Inheritance, which includes DMD. Thus, we propose that the loss of Matr3 destabilizes the chromatin loops at conserved hotspots related to muscular disease, and the rearrangement of loops contributes to changes in transcription factor bindings and downstream gene expression that leads to defects in the development.

Minor

1. Fig 1c: quantification is needed.

For quantification, we performed different assays to measure differentiation, myotube fusion, myotube density. Following an established protocol (Quinn et al. Nat Commun. 2017), we performed differentiation assays (fraction of nuclei contained within all MHC+ cells as compared with the number of total nuclei) (Supplemental Figure 3a) and the fusion assay (the fraction of nuclei contained within MHC + myotubes which had >3 nuclei, as compared to the number of total nuclei) (Supplemental Figure 3b) and quantified myotubes density in each condition (Figure 1c). We observed that myotube density (the myotube number in the same area across samples, at least 6 replicates) significantly increased in Matr3 knockout.

2. The conclusion “depletion of Matr3 leads to aberrant muscle differentiation” is overstated, as there is no in-vivo data to support “muscle differentiation”.

We have re-phrased the term to “myogenesis”, “cell development”

3. Fig 3a-b: Scales should be added; to show the difference, average profiles (as those

shown in Fig 4a) should be also presented. Fig 3c-d, gene symbols and heatmap cells are non-corresponding. Any statistical test was used for Fig3h-3i?

Done adding scale, adding average profiles, and fixing gene symbols non-corresponding issue.

We added p-value to Fig 3h-3i. The statistical test we performed was paired t-test.

4. Fig 4a and 5c-5e: Scales should be added.

Done adding scale bar to Figure 4a.

Figure 5c-3 already have color bars below the plots (see horizontal color scale bars)

5. From the data presented in Fig S7, it remained unclear that “chromatin loops were correlated with differential open chromatin accessibility. What the colors mean in Fig S7a? There is no clear trend in this figure.

We have remade the figure and rewrote the figure legend to improve clarity (Please see current version Supplement Figure 9). We interrogated the strength of Hi-C interactions for pairs of anchors overlapping with ATAC-seq peaks (termed ATAC anchor 1 and anchor 2) (see the cartoon on top of FigS 9a). For a pair of anchors (x, y), where x is anchor 1 and y is anchor 2, both of which overlap with ATAC-seq, we counted the number of differential Hi-C interactions (WT vs KO) happening at (x, y). ATAC anchors were sorted into 25 bins based on WT/KO ATAC-differential P-values of individual anchors (see axis titles). Thus, as result of this sorting, the top left corner of the heatmap (boxed) quantifies the number of differential Hi-C interactions between the most significant differential ATAC-seq peaks serving as anchors (refer to the color bar). Then the top 10% fraction (boxed in FigS 9a), divided into 10 X 1% bins (shown in FigS 9b). Each histogram quantified the number of differential Hi-C interactions observed in that percentile bin, showing the correlation between chromatin looping and differential chromatin accessibility.

6. The quality of Fig 2c should be improved.

We repeated the experiment in Figure 2c and provide an update figure.

Reference

Cha, Hye Ji et al. “Inner nuclear protein Matrin-3 coordinates cell differentiation by stabilizing chromatin architecture.” *Nature communications* vol. 12,1 6241. 29 Oct. 2021, doi:10.1038/s41467-021-26574-4

Martin, Eric W et al. “Chromatin accessibility maps provide evidence of multilineage gene priming in hematopoietic stem cells.” *Epigenetics & chromatin* vol. 14,1 2. 6 Jan. 2021, doi:10.1186/s13072-020-00377-1

Quinn, Malgorzata E et al. "Myomerger induces fusion of non-fusogenic cells and is required for skeletal muscle development." *Nature communications* vol. 8 15665. 1 Jun. 2017, doi:10.1038/ncomms15665

Wang, Bao et al. "SATB1 regulates 3D genome architecture in T cells by constraining chromatin interactions surrounding CTCF-binding sites." *Cell reports* vol. 42,4 (2023): 112323. doi:10.1016/j.celrep.2023.112323

Zhang, Yuwen et al. "MATR3-antisense LINE1 RNA meshwork scaffolds higher-order chromatin organization." *EMBO reports* vol. 24,8 (2023): e57550. doi:10.15252/embr.202357550

Zheng, Rong et al. "LINE-associated cryptic splicing induces dsRNA-mediated interferon response and tumor immunity." *bioRxiv : the preprint server for biology* 2023.02.23.529804. 24 Feb. 2023, doi:10.1101/2023.02.23.529804. Preprint.

REVIEWERS' COMMENTS

Reviewer #1 (Remarks to the Author):

Upon a thorough re-evaluation, I am pleased to note that the authors have diligently and effectively addressed the concerns I highlighted in my previous review. The revisions made have notably improved the quality of the manuscript, both in terms of its scientific rigor and the clarity of its presentation.

Reviewer #2 (Remarks to the Author):

In this revised manuscript, my previous questions have been addressed. It should be suitable for publication in Nature Communications.